



# Exploring temporal and spatial variation of nitrous oxide flux using several years of peatland forest automatic chamber data

Helena Rautakoski[1], Mika Korkiakoski[1], Jarmo Mäkelä[2], Markku Koskinen[3], Kari Minkkinen[4], Mika Aurela[1], Paavo Ojanen[4,5], Annalea Lohila[1,3]

[1]Finnish Meteorological Institute, P.O. Box 503, FI-00101 Helsinki, Finland
[2]Advanced Computing Facility, CSC – IT Center for Science Ltd, P.O. Box 405, FI-02101 Espoo, Finland
[3]Institute for Atmospheric and Earth System Research, University of Helsinki, Gustaf Hällströmin katu 2, P.O. Box 64, FI-00014 Helsinki, Finland
[4]Department of Forest Sciences, University of Helsinki, P.O. Box 27, FI-00014 Helsinki, Finland
[5]Natural Resources Institute Finland, Viikinkaari 4, FI-00790 Helsinki, Finland

*Correspondence to*: Helena Rautakoski (helena.rautakoski@fmi.fi)

**Abstract:** The urgent need to mitigate climate change has evoked a broad interest in better understanding and
estimating nitrous oxide ($N_2O$) emissions from different ecosystems. Part of the uncertainty in $N_2O$ emission estimates still comes from an inadequate understanding of the temporal and small-scale spatial variability of $N_2O$ fluxes. Using 4.5 years of $N_2O$ flux data collected in a drained peatland forest with six automated chambers, we explored temporal and small-scale spatial variability of $N_2O$ fluxes. A Random forest with conditional inference trees was used to find immediate and time-lagged relationships between $N_2O$ flux and environmental conditions
across seasons and years with different environmental conditions.

The temporal variation of $N_2O$ flux was large, and the daily mean flux varied between –11 and 1760 µg $N_2O$ m$^{-2}$ h$^{-1}$. Three of the six measurement chambers had a maximum $N_2O$ flux of less than 400 µg $N_2O$ m$^{-2}$ h$^{-1}$, while the fluxes in the other three chambers exceeded 1000 µg $N_2O$ m$^{-2}$ h$^{-1}$. Spatial differences in the flux persisted over time, and despite the high small-scale spatial variability, the temporal patterns of the fluxes were relatively
similar across the chambers. Soil moisture as well as air and soil surface temperature were the most important variables in the random forest, with lagged soil moisture also considered important. $N_2O$ flux responded to soil wetting with a time lag of 1–7 days, but the length of the time lag varied spatially and between seasons indicating interactions with other spatially and temporally variable environmental conditions.

The high temporal variation in $N_2O$ flux was related to a) seasonally variable environmental conditions,
with the highest $N_2O$ fluxes measured after summer dry-wet cycles and winter soil freezing, and b) to annually variable seasonal weather conditions, which lead to high year-to-year variability in $N_2O$ budget. Changes especially in the frequency of summer precipitation events and in winter temperature and snow conditions may increase the variability of annual $N_2O$ emissions if the variability in summer and winter weather conditions increases due to climate change.




## 1. Introduction

Among the greenhouse gases, whose emissions contribute to climate change, one of the most potent is nitrous oxide ($N_2O$), with a global warming potential 260 times stronger than carbon dioxide (Myhre et al., 2013). A major part of the emissions of $N_2O$ originates from soils (Butterbach-Bahl et al., 2013; Davidson and Kanter, 2014), and human impact through altered nitrogen (N) cycle, land use and climate change affect the soil $N_2O$ emissions both in natural and managed ecosystems (Tian et al., 2018, 2020). The urgent need to mitigate climate change has evoked a broad interest in better understanding and estimating $N_2O$ emissions of different ecosystems (Thompson et al., 2019; Shakoor et al., 2021). However, the accurate estimation of $N_2O$ emissions has remained a challenge and emissions estimates continue to have relatively high uncertainties (Tian et al., 2018, 2020). A large part of the uncertainty in $N_2O$ emission estimates comes from inadequate understanding of the temporal and small-scale spatial variability of $N_2O$ fluxes (Sutton et al., 2007; Groffman et al., 2009; Kuzyakov and Blagodatskaya, 2015; Wang et al., 2020).

$N_2O$ is formed in multiple processes, each favored by different soil conditions (Butterbach-Bahl et al., 2013). The main processes producing $N_2O$ in soils are nitrification and denitrification (Bollmann and Conrad, 1998; Zhu et al., 2013; Hu et al., 2015). Nitrifying bacteria turn ammonium into nitrate in aerobic conditions. Nitrate produced in nitrification can further be reduced to nitric oxide, $N_2O$ and gaseous nitrogen ($N_2$) in oxygen-limited or anaerobic conditions (Wrage et al., 2001; Zhu et al., 2013; Wrage-Mönnig et al., 2018), making oxygen content a key control of $N_2O$ flux (Song et al., 2019). Oxygen limitation in soil and substrate availability for microbes is affected by soil water content, which makes $N_2O$ production also sensitive to varying soil moisture conditions (Butterbach-Bahl et al., 2013). Along with soil moisture, substrate availability is widely affected by human actions, such as fertilization, nitrogen deposition and drainage of organic soils, which are all linked to increased $N_2O$ fluxes (Pärn et al., 2018; Tian et al., 2020; Lin et al., 2022). Soil temperature regulates microbial activity in the soil, but it also shapes microbial community composition and affects $N_2O$ production through, for example, frost, ice formation and thaw (Holtan-Hartwig et al., 2002; Risk et al., 2013; Wagner-Riddle et al., 2017).

Temporal variation of soil conditions and substrate availability can lead to a high temporal variation of $N_2O$ flux within a year (Groffman et al., 2009; Kuzyakov and Blagodatskaya, 2015). Soil freeze-thaw and dry-wet cycles are examples of changes in soil conditions shown to shape seasonal variation in $N_2O$ emissions (Risk et al., 2013; Congreves et al., 2018). High temporal variation has been shown to be typical for $N_2O$ flux in several ecosystems (Luo et al., 2012; Molodovskaya et al., 2012; Anthony and Silver et al., 2021), but understanding related to the temporal variation of $N_2O$ production is limited by sparse sampling intervals of manual flux measurements, lack of short-interval measurements and poor temporal coverage of data from all parts of the year (Barton et al., 2015; Grace et al., 2020). Since short periods of high $N_2O$ fluxes can account for a substantial amount of the annual $N_2O$ budget (Molodovskaya et al., 2012; Ju and Zhang, 2017; Anthony and Silver, 2021), capturing $N_2O$ flux peaks and understanding the causes of temporal variation of $N_2O$ flux are essential for estimating annual emissions accurately.

Similar to temporal variation, high spatial variation is common for $N_2O$ flux (Groffman et al., 2009). Estimating $N_2O$ emissions accurately requires integrating information about the temporal and spatial dynamics. Variation in $N_2O$ flux occurs on multiple spatial scales, from large-scale variation between ecosystems to small-



scale variation within a few meters (Ojanen et al., 2010; Krichels and Yang, 2019). High N$_2$O fluxes are typically measured in ecosystems with high N availability, such as in agricultural fields and in drained organic soils where

fertilization and organic matter mineralization provide N supply for N$_2$O production (Maljanen et al., 2003; Reay et al., 2012; Leppelt et al., 2014; Pärn et al., 2018). Within an ecosystem, varying soil properties and conditions such as organic matter content, soil moisture or pH can create spatial variability in the N$_2$O fluxes (Jungkunst et al., 2012; Giltrap et al., 2014). Although the small-scale spatial variation of N$_2$O flux can be large and exceed the spatial variation between more distant parts of the same ecosystem (Yanai et al., 2003; Jungkunst et al., 2012; Giltrap et al.,

2014), the causes of small-scale spatial variability of N$_2$O are poorly known and little studied, especially with short-interval measurements. Several questions related, for example, to the persistence of spatial patterns over time and linkages between the spatial and temporal variation of N$_2$O flux are little understood.

Drained peatland forests are examples of ecosystems with relatively high N$_2$O fluxes and high spatio-temporal variation of those fluxes (Maljanen et al., 2003; Ojanen et al., 2010; Pärn et al., 2018). In Finland about 60 %

of the original peatland area has been drained for forestry (Korhonen et al., 2021). The drainage has resulted in a lowered groundwater level and increased N availability from the decomposing peat, leading to increased N$_2$O fluxes, especially in nutrient-rich peatland forests with a low C:N ratio (Martikainen et al., 1993; Laine et al., 1996; Klemedtsson et al., 2005). The focus of previous studies on peatland forest N$_2$O fluxes has been on understanding the large-scale spatial variation of N$_2$O fluxes between peatland forests (Klemedtsson et al., 2005; Ojanen et al.,

2010; Minkkinen et al., 2020) and reporting N$_2$O fluxes for the studied peatland forest sites in response to harvesting (Huttunen et al., 2003; Korkiakoski et al., 2019, 2020). The temporal variation of N$_2$O flux as well as its linkages to smaller-scale spatial variation of the flux are not well understood, and only one snapshot of short-interval N$_2$O measurements is available from drained boreal peatland forest (Pihlatie et al., 2010).

For the first time in boreal peatlands and non-agricultural boreal ecosystems, we use several years (2015–

2019) of automated chamber measured N$_2$O flux to gain a more comprehensive understanding of the spatio-temporal dynamics of N$_2$O flux. We investigate the characteristics of temporal and small-scale spatial variation in N$_2$O flux and link the temporal variation of N$_2$O flux to seasonally and annually variable environmental conditions including immediate and time-lagged responses. This is done to form a more comprehensive understanding about the spatio-temporal dynamics of N$_2$O flux and to decrease uncertainties in current and future N$_2$O emission estimates in boreal

peatland forests and beyond.

## 2. Materials and methods

### 2.1. Site description

The measurements were made in 2015–2019 in Lettosuo, a drained nutrient-rich peatland forest located in

southern Finland (60°38′ N, 23°57′ E). The annual mean temperature in the area is 5.2 °C, and the mean annual precipitation is 621 mm according to long-term weather record from the nearest automatic weather station (Jokioinen Ilmala, 1991–2020). The site was first drained in the 1930s and more intensively in 1969 to enhance tree growth. The site was fertilized with phosphorus and potassium after the later drainage. The relatively low C:N ratio reflects the fen history of the site (Table 1). Ditches were dug in 1969 about 1 m deep with 45 m spacing. Drainage





lowered the groundwater level, resulting in a transition to boreal-forest-like vegetation. The ground vegetation consisted mainly of dwarf shrubs (*Vaccinium myrtillus, Vaccinium vitis-idaea*) and herbaceous plants (*Lysimachia europaea, Dryopteris carthusiana*) with sedges (*Carex globularis, Eriophorum vaginatum*) and Sphagnum mosses (*Sphagnum russowii, Sphagnum girgensohnii*) in patches. Before March 2016, the site was a mixed forest dominated by Scots pine (*Pinus sylvestris*) as an overstory, while the understory consisted of mostly Norway spruce (*Picea*

*abies*). Both over and understory included a small number of Downy birch (*Betula pubescens*). In March 2016, overstory pine trees were harvested (70 % of the total stem volume; Korkiakoski et al., 2020, 2023), but the surroundings of the measurement chamber used in this study were harvested more lightly. The study plots continued to have high coverage of spruce and birch after the overstorey pine trees were removed in the harvesting. The partial harvesting did not affect $N_2O$ fluxes according to the previous study from the site (Korkiakoski et al., 2020), and the

effect of the harvesting was left out of the focus of this study.

**Table 1: Soil properties at the study site. Values represent general soil properties at the study site before forest harvesting was done. Data from Korkiakoski et al. (2019).**


| Depth | Total-N (%) | Total-C (%) | C:N | Bulk density (g cm$^{-3}$) |
|---|---|---|---|---|
| Humus | $1.7 \pm 0.4$ | $56.2 \pm 2.3$ | $33.2 \pm 2.3$ | $0.01 \pm 0.003$ |
| 0–10 cm | $2.2 \pm 0.2$ | $55.2 \pm 2.1$ | $24.9 \pm 2.1$ | $0.12 \pm 0.03$ |
| 10–20 cm | $2.5 \pm 0.2$ | $58.9 \pm 1.6$ | $23.8 \pm 1.6$ | $0.18 \pm 0.02$ |

### 2.2. Automatic chamber measurements

$N_2O$ flux between the forest floor and atmosphere was measured with six automatically operating chambers. The transparent, acrylic, rectangular cuboid chambers with the dimensions 57 x 57 x 40 cm (length x width x height) were placed to sample the spatial variation of the ground vegetation composition and were located within an area of 15 x 20 m (Fig. 1). Distance to the closest ditch and trees also varied between chambers (Table S1). The chambers were placed on permanently installed steel collars that were inserted into the soil to about 2 cm depth. All the

chambers closed automatically for six minutes once an hour year-round resulting in 6 x 24 flux measurements per day. Chambers had temperature sensors measuring the headspace temperature and a fan to mix the air inside the chamber headspace. $N_2O$ concentration of the chamber headspace air was measured using a continuous-wave quantum cascade laser absorption spectrometer (LGR-CW-QCL $N_2O$/CO-23d, Los Gatos Research Inc., Mountain View, CA, USA) that was placed in the measurement cabin close to the chambers. The sample air was pumped into

the analyzer and back to the chamber headspace through plastic tubes (length 15 m). The same chamber measurement system was used also in other studies covering $N_2O$, $CO_2$ and $CH_4$ fluxes of the same site (Koskinen et al., 2014; Korkiakoski et al., 2017, 2020).

N$_2$O fluxes were calculated using a linear fit to the $N_2O$ concentration change during the chamber closure. Calculated fluxes were filtered using normalized root mean square error threshold and iterative standard deviation

filter to remove erroneous fluxes resulting from chamber malfunction. A more detailed description of the flux



calculation and filtering can be found in Korkiakoski et al., (2020). The fact that the fans were not adjusted according to the wind conditions likely created some diurnal cycle in the flux, as discussed previously for $CO_2$ and $CH_4$ fluxes at the same site (Koskinen et al., 2014; Korkiakoski et al., 2017). To minimize the possible effect of artificial diurnal variation in $N_2O$ flux, daily mean fluxes were used in this study.


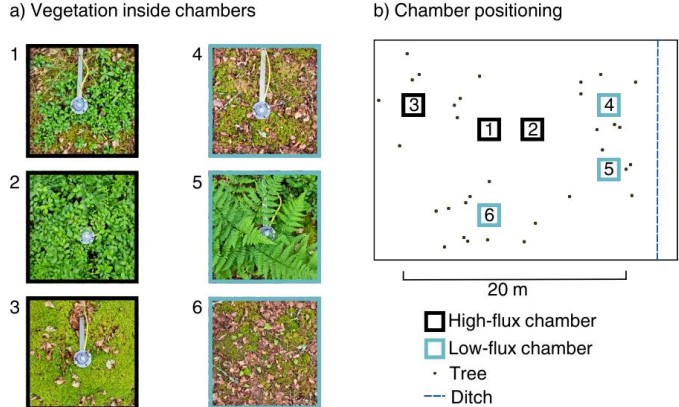

**Figure 1: a) Vegetation inside the six chambers and b) the positioning of the chambers on the forest floor in relation to the nearest ditch and trees. Chambers are named from one to six based on the maximum measured flux with Chamber 1 having the highest measured flux. Chambers 1–3 with black edges are classified as high-flux chambers and Chambers 4–6 with blue edges as low-flux chambers (Sect. 3.2).**


### 2.3. Environmental variables

Several environmental variables were measured to link the temporal variation of $N_2O$ fluxes with the environmental conditions. Air temperature was measured at 2 m height below the forest canopy (HMP45D, Vaisala

Oyj, Vantaa, Finland). Soil surface temperature was measured at 2 cm depth in each chamber and the soil temperature at 5 cm depth at one location close to the chambers (Pt100, Nokeval Oy, Nokia, Finland). Soil moisture was measured at one location about 75 m from the chamber measurement location at 7 and 20 cm depths (Delta-T ML3, Delta-T Devices Ltd, Cambridge, UK). The soil moisture data were used to describe the temporal variation of soil moisture rather than the absolute level of soil moisture at the chamber location. We assumed that the measured

soil moisture conditions represent the conditions near the chambers relatively well since the microtopography, surface vegetation and shading by the canopy were relatively similar in both locations.

Water table level (WTL) below the soil surface was measured hourly using automatic probes (TruTrack WT-HR, Intech Instruments Ltd, Auckland, New Zealand; Odyssey Capacitance Water Level Logger, Dataflow Systems Ltd, Christchurch, New Zealand) placed into dipwells that were installed into the ground. Chambers 1–2

and 3–4 shared a WTL sensor that was placed in between the two chambers, and Chambers 3 and 6 had their own WTL sensors next to the chamber collar. Since WTL measurements for Chambers 3, 4, 5 and 6 started in December 2015, WTL before that was modeled for each chamber using Random forest with conditional inference trees (Hothorn et al., 2006). Other WTL measurements near the automatic chambers, precipitation and soil moisture were used as explanatory variables in the models (evaluation data $R^2 = 0.90$–$0.97$).



Precipitation was measured at the site (Casella Tipping Bucket Rain Gauge, Casella Solutions Ltd, Bedford, UK; OTT Pluvio2 L 400 RH, OTT Hydromet Ltd, Kempten, Germany) and daily cumulative precipitation used. The precipitation data measured in the nearest weather station was used to gap-fill winters and other measurement gaps in precipitation data measured at the site (correlation of precipitation between sites 0.65, $p <$ 0.05). Snow depth was measured in the nearest weather station and used to describe general snow conditions

experienced each winter.

Thermal seasons were used to analyze the seasonality of $N_2O$ fluxes. The thermal seasons were defined according to typical Finnish standards (Ruosteenoja et al., 2011; Finnish Meteorological Institute, 2023), and by using air temperature data of the site (Appendix A). During thermal winter, daily mean air temperature was persistently below 0 °C, during summer above 10 °C and during thermal spring and autumn between 0 and 10 °C.

Seasons based on months are used to compare conditions measured at the site with seasonal long-term averages reported for the nearest automatic weather station.

### 2.4. Identifying high-flux periods

The term high-flux period was used to describe periods of elevated flux, including periods from moderately

increased flux to the highest flux peaks. The term high-flux period was used instead of a commonly used hot moment term because the definition of a hot moment largely varies between studies, with sometimes only extremely high fluxes being considered as hot moments (Molodovskaya, 2012; Krichels et al., 2019; Anthony and Silver, 2021; Song et al., 2022).

To identify high-flux periods and to numerically describe the temporal patterns of $N_2O$ fluxes, different

thresholds to separate high-flux days from the baseline days were tested. Finally, a common percentile threshold of 70 % was used in all chambers. High fluxes were measured less frequently compared to the more common low fluxes, which made high-flux days distinct from the more common baseline days in flux histograms of all chambers (Fig. S2). Any percentile threshold between 60–80 % separated high-flux days from the baseline relatively well, and the mean of these (70 %) was used. The mean $N_2O$ flux of the study period was close to the chosen 70 % percentile

threshold in all chambers. Days with the mean flux above the 70 % percentile were classified as high-flux days.

The length of each high-flux period was the number of days the flux remained above the 70 % percentile, including possible data gaps within this period. The high-flux period was set to continue over the data gap if three days before and after the data gap were classified as high-flux days. A three-day marginal was chosen to ensure that short one-to-two-day peaks would not create long-lasting high-flux periods over the data gaps. If the high-flux

period started from a data gap or ended to it, the start or end date of the high-flux period was set to the first or last measured day, respectively.

Pearson correlation was used to test correlation between N2O flux time series of different chambers and multiple linear regression was used to test if each environmental variable could explain differences in the flux patterns between chambers. In the multiple linear regression, N2O flux of each chamber was explained by flux of

one other chamber, and ability of each environmental variable to explain the remaining variance was tested one environmental variable at the time.



### 2.5. Machine learning

Machine learning models were used to improve understanding of the temporal controls on $N_2O$ flux, including a possible effect of time lags between environmental conditions and $N_2O$. Since the models were run separately for the six chambers, the models also allowed estimation of whether the temporal variation is controlled similarly in the different chambers. The machine learning approach was used because machine learning models do not rely on mathematical functions to describe relationships between variables and are able to account for interactions between variables without having to include them in the equation by hand (Olden et al., 2008). This is particularly useful when using a large dataset with multiple environmental variables to model $N_2O$ fluxes whose controls and mathematical forms of responses are not yet fully understood.

The Random forest algorithm, developed by Breiman (2001), is a classification tree-based method that uses bootstrap aggregation of a model training data and a randomly chosen subset of explanatory variables (*mtry*) to train each classification tree. In bootstrap aggregation, a subset of data is taken from the model training data with or without returning it to the original training data. The part of data that is not bootstrapped to train trees is called out-of-bag (OOB). OOB data can be used to evaluate model performance since this part of the data is not used during the model training phase. In each Random forest tree, the bootstrapped data are classified into subgroups and further to smaller subgroups by setting threshold values for the randomly chosen subset of explanatory variables. The setting of the threshold values is done to maximize the information gain until no further thresholds, also called splits, can be made. After a selected number of trees are built, the final model prediction can be made using the average of all the trees (continuous response) or the most common outcome (categorical response).

Random forest variable importance (VI) metrics show the importance of each explanatory variable in explaining variation in the response variable. Variable importance metrics can be biased if the data type and scale of the explanatory variables vary or if there is a correlation between explanatory variables (Strobl et al., 2007). Therefore, we used Random forest with conditional inference trees (Hothorn 2006) that allowed us to get more accurate variable importance measures in the presence of correlated explanatory variables and their time-lagged versions. Compared to trees in Random forest, conditional inference trees use a p-value-based splitting criterion to classify the bootstrap aggregated data in the building phase of each tree. As suggested by Strobl et al. (2007), in the presence of correlated explanatory variables, variable importance metrics from the conditional inference trees were calculated using conditional permutation importance.

Chamber-specific models had $N_2O$ flux as the response variable and the measured temperature variables (air, 2, 5 cm), soil moisture (7 and 20 cm), WTL and daily cumulative precipitation as explanatory variables. Time lags of 1–7 days were added as additional explanatory variables for all the explanatory variables. The imbalanced distribution of $N_2O$ fluxes as model predictors were corrected with the SMOGN algorithm (Abd Elrahman and Abraham, 2013). The subset of data to train each tree was bootstrapped without replacement with a sample size 0.632 times the size of the training dataset, as suggested by Strobl (2007). Models were trained with 500 trees and Random forest default *mtry* for continuous response variable was used (*mtry* = number of explanatory variables / 3).



The first three years of data were utilized as the model training period (1 June 2015–1 June 2018), and this data were further split into 70 % training data and 30 % evaluation data to test model performance within the training period. The fourth year of measurements until soil moisture measurements ended (1 June 2018–4 April 2019) was left aside for evaluation to test model performance outside the training period. The prediction accuracy of the models in each evaluation data was analyzed using R squared ($R^2$) and root mean squared error (RMSE). Evaluation results are presented in appendices (Appendix B). Variable importance values were scaled between zero and one to enable comparison between chambers. The Accumulated local effects (ALE) method by Apley and Zhu (2020) was used to visualize the response of $N_2O$ flux to environmental conditions and their lags.

### 2.6. Gap-filling and $N_2O$ budgets

Data gaps covered 12–24 % of the study period depending on the chamber. Most gaps occurred at the same time in all chambers. Notable is that measurements in Chamber 6 ended six months earlier in 2019 than measurements in other chambers. $N_2O$ flux time series were gap-filled to calculate $N_2O$ budgets. In other analysis, gap-filled data were not used to avoid additional uncertainty of the results arising from the gap-filling.

Gap-filling was done by training the Random forest with conditional inference trees on the whole measurement period (4.5 years) data with 30 % data excluded for evaluation. The same explanatory variables were used in the models as in the analysis, including time-lagged variables. Evaluation results of gap-filling models are shown in Appendices (Appendix B). Gap-filled daily mean $N_2O$ fluxes were used to calculate cumulative $N_2O$ flux for each chamber in each thermal season and year. The uncertainties related to the $N_2O$ budgets were assumed to be a combination of uncertainty related to flux measurement and uncertainty related to gap-filling. Detailed information about the calculation of the uncertainty can be found in Korkiakoski et al. (2017).

Flux calculation was performed in the Python programming language version 2.7 (Van Rossum and Drake, 1995). Data preparation and analysis were performed in R statistical software version 1.4.1 (R core team, 2021). Cforest in the party package (Hothorn et al., 2006; Strobl et al., 2007; Zeileis et al., 2008) was used for Random forest with conditional inference trees.

## 3. Results

### 3.1. Environmental conditions

The seasonal temperature conditions were variable for the years 2015–2019 (Fig. 2). The summers (June, July, August) 2015 (14.1 °C) and 2017 (14.4 °C) were colder than the long-term average (15.6 °C), while winters (December, January, February) 2015–2016 (–3.4 °C), 2016–2017 (–3 °C) and 2018-2019 (–3.5 °C) were warmer than the long-term average (–4.3 °C) (Jokioinen Ilmala, 1991–2020). Temperatures in all seasons during the years 2018 and 2019 were warmer than the long-term average, with summer (17.2 °C) and autumn (6.7 °C) 2018 being especially warm (long-term average temperatures 15.6 °C and 5.4 °C, respectively).

The area received the least amount of precipitation in 2018 (434 mm) and the most precipitation in 2017 (657 mm), when the long-term annual average was 621 mm. Winter 2015–2016 (67 mm) was wet, while autumn





2016 (36 mm), winter 2016–2017 (24 mm) and summer 2018 (44 mm) were dry compared to the long-term averages

(winter 44 mm, autumn 58 mm and summer 71 mm).

Soil conditions measured at the site varied between seasons and years (Fig. 3). Soil moisture at 7 cm was on average lower in winters (0.26 m$^{-3}$ m$^{-3}$) and springs (0.22 m$^{-3}$ m$^{-3}$) compared to summers (0.31 m$^{-3}$ m$^{-3}$) and autumns (0.33 m$^{-3}$ m$^{-3}$). Soil moistures at 7 cm and 20 cm were continuously lower than the means of the measurement period (0.28 and 0.56 m$^{-3}$ m$^{-3}$, respectively) from the summer 2018 until the end of the measurement

period. WTL was deeper than the mean of the study period (–36 cm) in the summer and autumn 2015 as well as in the summers 2018 and 2019. Soil surface temperatures varied on average between –0.6 °C in winter and 14.0 °C in summer with small differences in soil surface temperatures between chambers. Soil temperatures at 5 cm reached below zero temperatures in winters 2015–2016 (min –3.8 °C), 2016 –2017 (min –1.8 °C) and 2017–2018 (min –0.33 °C) with most days with negative soil 5 cm temperatures in winters 2015–2016 and 2016–2017. Temporal variation

in air and soil temperatures was greater in winters 2015–2016 and 2016–2017 compared to the latter two years of the measurement period. All winters had a period or periods of snow cover with the maximum measured snow depth being the greatest in winter 2018–2019 (52 cm) and the lowest in winter 2016–2017 (11 cm). Winters 2015–2016 (85 days) and 2016–2017 (93 days) had days with snow cover less than winters 2017–2018 (125 days) and 2018– 2019 (116 days).

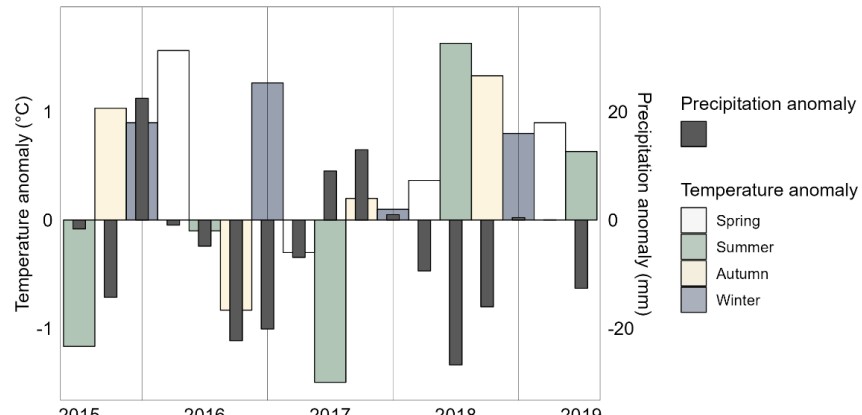

**Figure 2: Seasonal temperature and precipitation anomalies during the measurement period. The seasonal mean air temperature and seasonal cumulative precipitation of each year is compared to the long-term seasonal averages at the nearest weather station (1991–2020). Seasons are based on months (autumn: September–November, winter: December–February, spring: March–May and summer: June–August).**

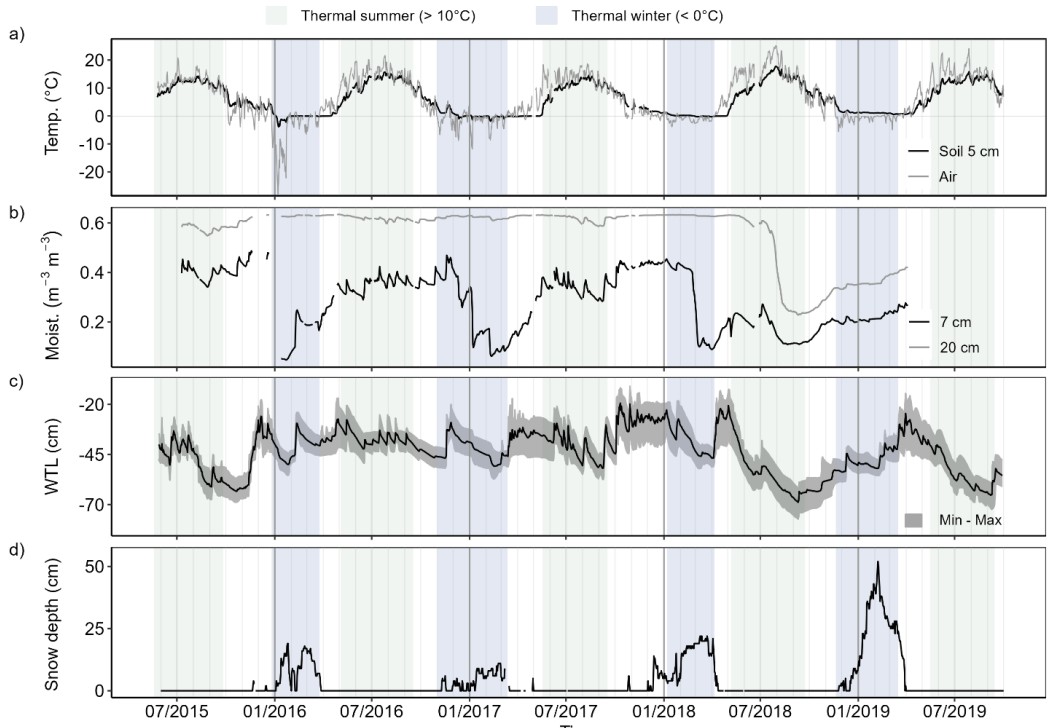

**Figure 3: (a) Daily mean air and soil temperatures (5 cm depth), (b) soil moisture (7 and 20 cm depth), (c) water table level (WTL) and (d) snow depth. WTL is the mean of the values measured next to the different chambers with variation between the lowest and highest WTL indicated with shading. Snow depth was measured at the nearest weather station. Data are not gap-filled. For the definition of thermal winter and summer, see Sect. 2.3).**

## 3.2. Temporal and spatial variation of $N_2O$ flux

Daily mean $N_2O$ flux varied between –10 and +1760 µg $N_2O$ m$^{-2}$ h$^{-1}$ during the 4.5 years of measurements (Fig. 4), and chamber mean $N_2O$ flux between +20 (Chamber 6) and +140 µg $N_2O$ m$^{-2}$ h$^{-1}$ (Chamber 1) (Table 2). The annual mean flux was the highest in 2016 or 2017, depending on the chamber, and smallest in 2018 in all chambers (Table S3.1). Mean fluxes in 2015 (June–December) were lower than in the whole years of 2016 and 2017 but higher than in 2018. Mean fluxes in 2019 (January–September) were generally higher than the mean fluxes in the whole year 2018.

Three chambers (Chambers 1, 2 and 3) had maximum daily mean fluxes larger than 1100 µg $N_2O$ m$^{-2}$ h$^{-1}$, while the other three chambers (Chambers 4, 5 and 6) had maximum daily fluxes smaller than 400 µg $N_2O$ m$^{-2}$ h$^{-1}$. The mean and the range of the daily mean $N_2O$ fluxes varied between years and chambers, but the high-flux chambers generally had a range and mean flux higher than the low-flux chambers in all years (Table S3.1). Differences in the mean and the range of the mean daily flux between high-flux and low-flux chambers were the largest in 2016 and 2017 and the smallest in 2018 and 2019. Based on the differences especially in the maximum



fluxes and in the range of the flux variation, Chambers 1–3 were classified as high-flux chambers and Chambers 4–6 as low-flux chambers.

Chamber-specific 70 % percentiles that were used to define high-flux periods from the baseline periods (Sect. 2.4) ranged from 20 to 170 µg $N_2O$ m$^{-2}$ h$^{-1}$ (Table 2). The length of the individual baseline periods varied between 1 and 330 days with a mean of 26 days, while the length of the high-flux periods varied between 1 and 134 days with a mean of 11 days.

        The correlations of the flux time series for each pair of chambers were positive and varied between 0.79

(Chambers 1 and 2) and 0.29 (Chambers 1 and 4) (Table S4.1). Correlations were the highest between the chambers with a similar range of $N_2O$ flux: among high-flux chambers, correlations varied between 0.64–0.79 and among low-flux chambers, between 0.46–0.49. Soil surface and soil 5 cm temperatures explained the differences in N2O fluxes between most chamber pairs statistically significantly (Fig. S4.2).

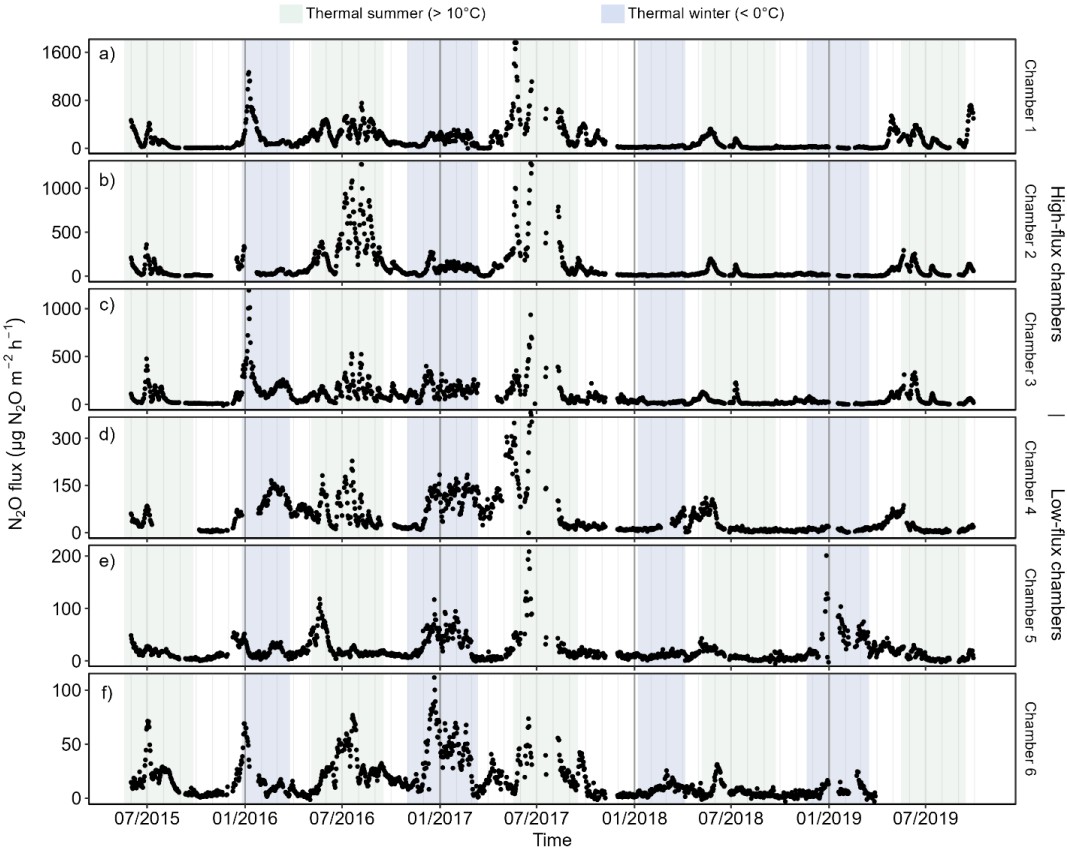

**Figure 4: Daily mean N₂O flux measured in the six automatic chambers in 2015–2019. Fluxes from different chambers are shown in panels (a–f) ordered by maximum daily mean N₂O flux. Chambers are grouped into low-flux (Chamber 1, 2 and 3) and high-flux chambers (Chamber 4, 5 and 6). The scale of the y-axis is chamber specific and fluxes are not gap-filled. Thermal winter refers to a period with daily mean air**



temperature persistently < 0 °C and thermal summer to a period with daily mean air temperature persistently > 10 °C.

**Table 2: Minimum, maximum, mean, median, 70 % percentile and standard deviation of daily mean N$_2$O fluxes over 4.5 years for each chamber. Unit of the flux is µg N$_2$O m$^{-2}$ h$^{-1}$. Percentile thresholds (70 %) were used to define high-flux periods.**

| Source | Min | Max | Mean | Median | Percentile 70 % | SD |
|---|---|---|---|---|---|---|
| Chamber 1 | −1 | 1761 | 143 | 73 | 168 | 193 |
| Chamber 2 | −1 | 1282 | 99 | 34 | 88 | 171 |
| Chamber 3 | −12 | 1192 | 87 | 46 | 100 | 112 |
| Chamber 4 | −1 | 381 | 48 | 22 | 58 | 57 |
| Chamber 5 | −5 | 244 | 20 | 13 | 20 | 23 |
| Chamber 6 | −3 | 112 | 17 | 11 | 19 | 17 |

### 3.3. Seasonality of N$_2$O flux

The highest daily fluxes were measured during the thermal summers (Chambers 1, 2, 4 and 5) or winters (Chambers 3 and 6) depending on the chamber. The mean seasonal N$_2$O fluxes calculated for thermal seasons were also the highest for the thermal summers or winters throughout the study period. The mean N$_2$O flux was the smallest in autumn in all years and chambers. The percentage of measurement days identified as high-flux days was on average 24 % in spring, 38 % in summer and 44 % in winter, while the thermal autumns had 9 % days identified as high-flux days (Fig. 5). The proportion of high-flux days in each season varied between years with the highest proportions of winter high-flux days measured in 2015 and 2017, and the highest proportions of summer high-flux days measured in summers 2016 and 2017. Variation in the percentage of high-flux days between chambers was greatest for thermal winters.

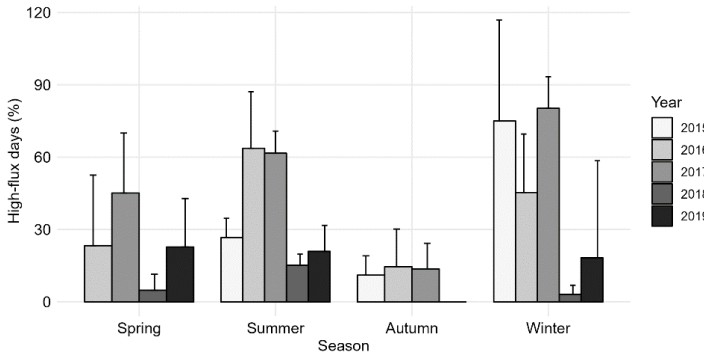

**Figure 5: Mean occurrence of high-flux days out of measured days in different thermal seasons and standard deviation between chambers.**



In spring, $N_2O$ fluxes started to increase when soil surface temperature increased above zero (Fig. 6 and S5), with most of the spring high-flux periods starting at soil surface temperatures 0–2 °C (Fig. 7). Spring $N_2O$ fluxes steadily increased with increasing soil temperatures, and flux peak top was reached in late spring or early summer. Increased summer $N_2O$ fluxes were measured after peaks in soil moisture and WTL, the highest $N_2O$ fluxes being reached typically several days after soil moisture and WTL peak. Most summer high-flux periods started when soil moisture at 7 cm was 0.37–0.41 $m^{-3}$ $m^{-3}$ and WTL between –35 and –50 cm. Autumn high-flux period starting conditions for soil moisture and WTL were similar to the summer, but increased $N_2O$ fluxes were reached a longer period of time after soil moisture and WTL peak. Soil temperatures at the start of the high-flux periods were lower in autumn compared to summer.

In early winter, the $N_2O$ fluxes increased when soil temperatures at the soil surface and 5 cm depth decreased close to zero and below that, with further increase in flux measured if soil temperature also at 5 cm depth decreased below zero (Fig. 6, 7, S5). After the initial freezing peak, early winter $N_2O$ flux started to decrease after the soil temperatures increased close to or above zero. Later during the winter, increased $N_2O$ fluxes were measured during periods of soil freezing or when soil temperatures increased close to or above zero after soil freezing. Freezing of the soil surface did not typically lead to high $N_2O$ fluxes without temperatures being below zero also at 5 cm. An exception to that was Chamber 5 during winter 2018–2019, where high $N_2O$ fluxes were measured during the mid-winter despite freezing temperature measured only at the surface soil. Temporal variation of $N_2O$ fluxes within winter were also related to the temporal variation in soil surface and air temperature, with $N_2O$ fluxes varying more in winters 2015–2016 and 2016–2017 with higher temporal variation in temperature compared to other winters.



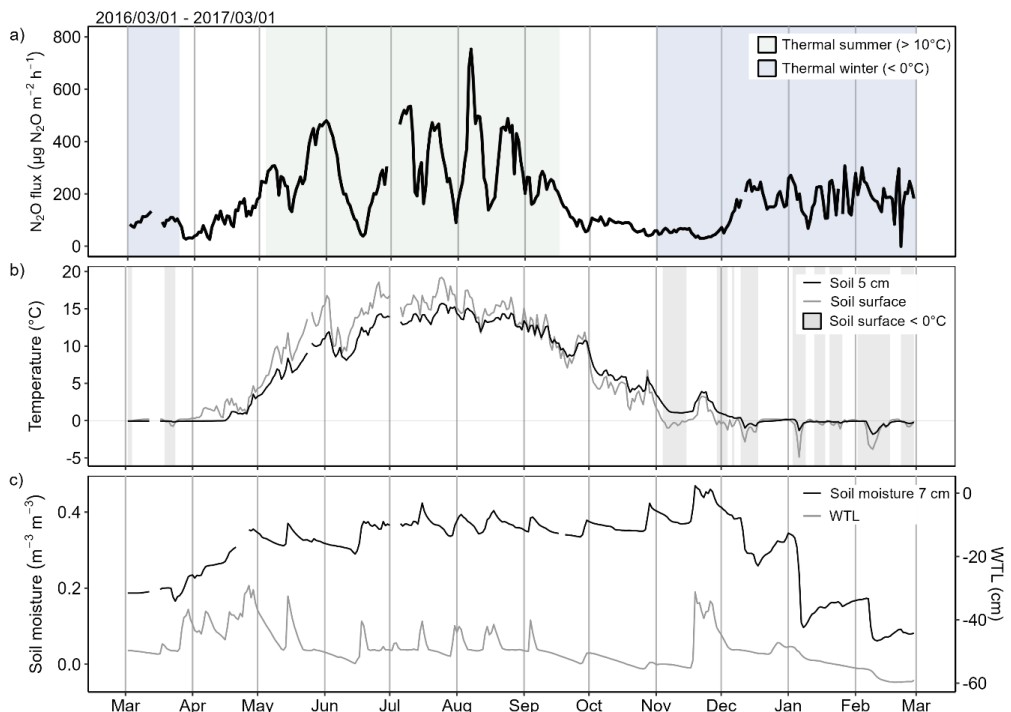

**Figure 6: a) Daily mean N₂O flux, b) soil surface temperature and temperature at 5 cm depth with highlighted freezing periods (soil surface temperature < 0°C), and c) soil moisture and water table level (WTL) from February 2016 to March 2017 in Chamber 1. The temporal variation of N₂O flux in Chamber 1 was similar to other chambers, but the range of flux variation was larger compared to low-flux chambers. The shown temporal dynamics of N₂O flux were measured in a year with relatively wet summer and warm winter. Data are not gap-filled. Figures for other chambers are presented in the supplements (S5).**




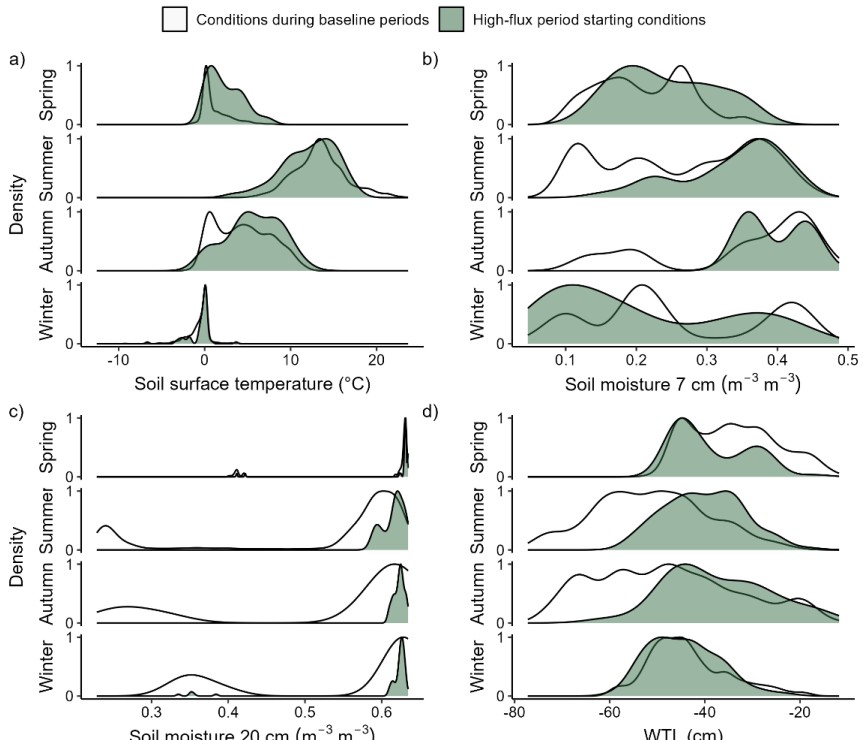

**Figure 7: Seasonal density distributions of high-flux periods starting in different a) soil surface temperatures, (b) soil moistures at 7 cm depth, (c) soil moistures at 20 cm depth, (d) and water table levels (WTL). Panels in each plot show density distribution for each thermal season. For comparison, the variation in soil conditions during baseline periods are also shown. All years and high-flux periods of all chambers are included. Density distribution values on y-axis are scaled (0−1).**

### 3.4. Modelling results

Unlagged soil moistures at 7 cm and 20 cm had mean variable importance (VI) scores 0.43 and 0.45 (respectively, 0 = lowest importance, 1 = highest importance) when VI scores were averaged across chambers (Fig. 8). Lagged (1−7 days) soil moisture variables received on average VI score of 0.31 (7 cm soil moisture) and 0.33 (20 cm soil moisture). The average VI score for unlagged air temperature was 0.45 and for soil surface temperature 0.24 and the variable importance generally decreased with increasing lag time. Unlagged soil temperature at 5 cm received VI score of an average 0.27 and increased VI scores also for lagged variables with the mean across lags 0.25. VI scores for WTL were on average 0.04 with little importance for lagged WTL in most chambers. Precipitation received VI score of 0.06 and increasing importance with increasing lag time. The most important variable and the importance of their individual lags varied between chambers with either soil moisture, WTL or air temperature receiving the highest VI score. High-flux chambers received high VI scores also for lagged temperature variables that were less important in low-flux chambers.



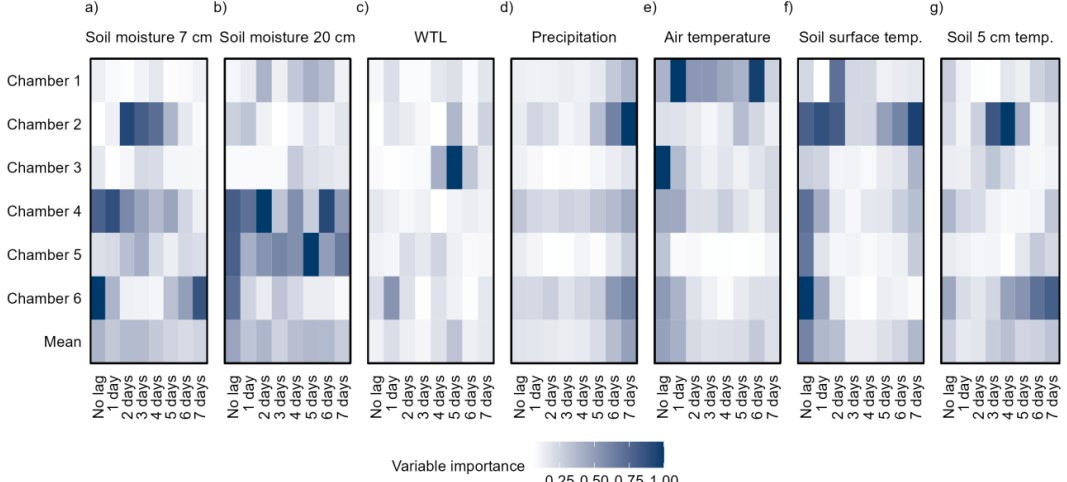

**Figure 8: Variable importance (VI) scores of different environmental variables and their lagged versions in explaining the temporal variation of N$_2$O. The matrix plot shows VI values separately for different chambers (Chambers 1–6) as well as the mean VI across all the chambers (Mean). VI values are means across 10 runs of Random forest with conditional inference trees. VI scores are scaled between 0 and 1 (0 = lowest importance, 1 = highest importance) per chamber to make VI scores comparable across chambers.**

ALE curves for unlagged 7 cm soil moisture showed the highest N$_2$O fluxes generally predicted on soil moisture values close to 0.3 m$^{-3}$ m$^{-3}$ or below 0.1 m$^{-3}$ m$^{-3}$ (Fig. 9 and S6). On moist conditions (> 0.3 m$^{-3}$ m$^{-3}$), the highest fluxes were predicted for lagged soil moisture. Predicted flux was typically the highest if soil moisture had been greater than 0.4 m$^{-3}$ m$^{-3}$ 3–7 days ago. On low 7 cm soil moistures (< 0.3 m$^{-3}$ m$^{-3}$), the highest N$_2$O fluxes were predicted for unlagged soil moisture with little differences between chambers. On high 20 cm soil moisture (> 0.6 m$^{-3}$ m$^{-3}$), predicted N$_2$O flux increased with increasing soil moisture in all chambers. Predicted flux on high 20 cm soil moisture was the highest for lagged soil moisture only in two chambers. For WTL, the predicted flux was the highest when WTL had been closer to the soil surface than –30 cm 3–7 days ago. The highest flux for unlagged WTL was typically predicted for WTL deeper than –50 cm. In all chambers, N$_2$O flux was predicted to be the highest for 4–7 days lagged precipitation if rainfall had been about 5 mm or more.

On temperatures above 5°C, the predicted N$_2$O fluxes increased with increasing air and soil surface temperatures with the highest predicted fluxes taking place when air and soil temperature exceeded 15 °C and 10 °C, respectively. Below about 0–2 °C temperatures, the predicted N$_2$O fluxes increased with decreasing air, soil surface and soil 5 cm temperature. In most chambers, increase in the predicted flux on soil 5 cm temperature at 0–2°C was especially strong with differences in the responses between immediate and lagged variables between chambers. Responses between lagged and unlagged soil surface and air temperature variables also varied between chambers.




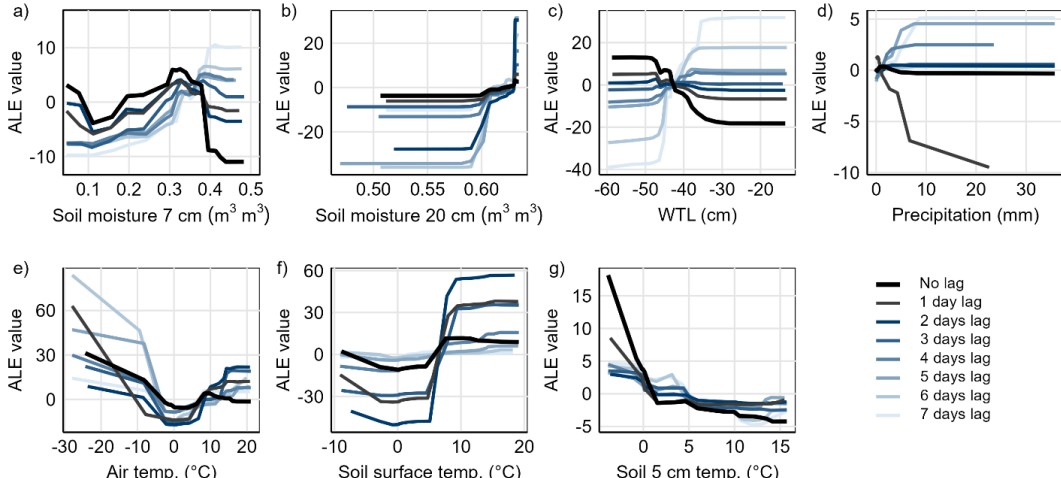

**Figure 9: Response curves between N₂O flux and environmental variables for Chamber 1 visualized using Accumulated Local Effects (ALE). Figures illustrate how the predicted N₂O flux values deviate from the mean predicted flux (ALE = 0) along the gradients of a) soil moisture at 7 cm depth, b) soil moisture at 20 cm depth, c) water table level (WTL), d) precipitation, e) air temperature, f) soil surface temperature and g) soil temperature at 5 cm. ALE responses for unlagged and lagged variables (1–7 days) are included. Lines represent the mean ALE values of 10 model runs. ALE responses for Chambers 2–6 are presented in supplements (S6).**

### 3.5. N₂O budgets

Annual N₂O budgets varied between 60 and 2110 mg N₂O m⁻² y⁻¹ when considering the three full measurement years 2016, 2017 and 2018 (Fig. 10, Tables S7). Annual N₂O budgets were higher than 1000 mg N₂O m⁻² y⁻¹ in high-flux chambers (Chambers 1–3) in 2016 and 2017, but less than 500 mg N₂O m⁻² y⁻¹ in all chambers in 2018. Winters and summers contributed generally the most to the annual N₂O budgets in all three years, with summers contributing on average 48 % and winters 34 % (Tables S7). The seasonal contributions to the annual N₂O budgets were, on average, 9 % for spring and autumn. Summer N₂O budgets in partially measured years 2015 and 2019 were smaller than in 2016 and 2017 but, especially in high-flux chambers, greater than in 2018.



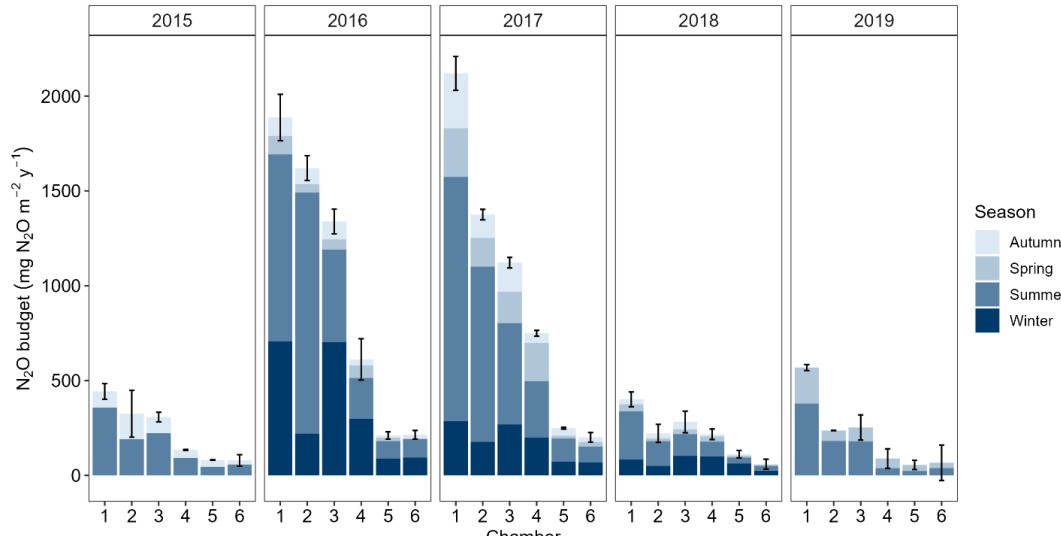

**Figure 10: Annual N$_2$O budgets for each chamber and measurement year with seasonal contributions. Only seasons that were completely within the measurement period (4.5 years) are included. The N$_2$O budget for the year 2015 only includes summer and autumn, and the N$_2$O budget for the year 2019 only spring and summer. Thermal seasons are used. Error bars denote total uncertainty related to the total N$_2$O budget of the year.**

## 4. Discussion

### 4.1. Temporal variation of N$_2$O fluxes

The measured peatland forest N$_2$O fluxes were relatively high compared to N$_2$O fluxes from most of the other boreal and temperate forests on peat or mineral soils. The N$_2$O budgets of boreal peatland forests have mainly varied between -30 and 920 mg N$_2$O m$^{-2}$ y$^{-1}$ (Alm et al., 1999; Arnold et al., 2005; Minkkinen et al, 2020; Butlers et al., 2023). The N$_2$O budgets of temperate mineral soil forests have varied within a similar range (Papen and Butterbach-Bahl, 1999; Luo et al., 2012). While the annual N$_2$O budgets in the present study were below 500 mg N$_2$O m$^{-2}$ y$^{-1}$ in 2018 in all measurement chambers, the annual N$_2$O budgets for three of the chambers exceeded 1000 mg N$_2$O m$^{-2}$ y$^{-1}$ in two years (2016 and 2017) out of the three full measurement years (Fig. 10). Similarly high or higher fluxes have been previously measured in peatland forest after clear-felling of the trees with especially logging residues linked with increased N$_2$O fluxes (Mäkiranta et al., 2012; Korkiakoski et al., 2019).

Nutrient-rich peat with relatively low C:N ratio likely explains high N$_2$O budgets of the study site. Low C:N ratio may have also increased sensitivity of N$_2$O fluxes to temporal variation in soil conditions (Klemedtsson et al., 2005; Pihlatie et al., 2010; Hu et al., 2015). Although the partial harvesting done at the site in spring 2016 did not increase N$_2$O budget of the harvested area compared to the control site according to Korkiakoski et al., (2020), the effect of harvesting on N$_2$O fluxes of individual chambers cannot be completely excluded. Since N$_2$O budgets increased after harvesting both in harvested site and in the control (see Korkiakoski et al., 2020), most of the



increase in $N_2O$ budgets in the years 2016 and 2017 is likely explained by year-to-year variation in environmental conditions.

Considering nutrient-rich soil and the tendency for high temporal variation of $N_2O$ flux in several ecosystems (Maljanen et al., 2010; Luo et al., 2012; Molodovskaya et al., 2012; Anthony and Silver, 2021), the complex temporal dynamics of $N_2O$ fluxes within and between years were expected. The high-flux period starting conditions and modelling results support previous evidence on the importance of freeze-thaw and dry-wet cycles strongly impacting temporal variation of $N_2O$ fluxes (Butterbach-Bahl et al., 2013; Risk et al., 2013; Wagner-Riddle

et al., 2017; Congreves et al., 2018). However, when comparing the temporal dynamics of $N_2O$ flux with those previously published from boreal and temperate regions (Maljanen et al., 2010; Pihlatie et al., 2010; Luo et al., 2012; Molodovskaya et al., 2012; Anthony and Silver, 2021; Gerin et al., 2023), the present data underline the importance of summer and winter $N_2O$ fluxes contributing to the annual $N_2O$ budget more than fluxes in spring. Previously, several studies on both peat and mineral soils have emphasized the importance of spring $N_2O$ fluxes in

the annual $N_2O$ budget, with distinct spring $N_2O$ peaks in some cases accounting for a large fraction of the annual budget (Pihlatie et al., 2010; Luo et al., 2012; Wang et al., 2023). In the present study, only moderately high $N_2O$ flux peaks were measured in spring and early spring $N_2O$ peaks were not typical. Year-to-year variation in $N_2O$ budgets was more attributed to variation in winter and summer $N_2O$ fluxes than variation in spring $N_2O$ fluxes.

During winters with discontinuous and shallow snow cover combined with high temporal variation in air

temperature below and above zero (2015–2016 and 2016–2017), the $N_2O$ fluxes were higher compared to the snowier winters with more stable temperature conditions (2017–2018 and 2018–2019). The insulating properties of the thicker snowpack may have prevented the soil from freezing to deeper depth, decreasing $N_2O$ fluxes during winter (Maljanen et al., 2009; Ruan and Robertson, 2017). Thicker snowpack combined with less variable air temperature conditions in the last two winters of the study period have likely decreased the number of freeze-thaw

cycles and decreased intensity of them leading to smaller total $N_2O$ flux during winter.

High-flux periods, especially during summers were linked with precipitation events that increased soil moisture and WTL. These high-flux events increased the total $N_2O$ budget of the rainy summers (2016 and 2017), while $N_2O$ budget in dry summer (2018) was low. Precipitation events may have increased the number of anoxic microsites in the soil favoring $N_2O$ production also through denitrification (Congreves et al., 2019; Song et al.,

2022). Active peat decomposition in the warm soil during the summer has likely also decreased oxygen availability in the soil and increased N availability from the mineralizing peat leading to high $N_2O$ emissions after summer rain events (Maljanen et al., 2003). Low surface soil moisture has likely limited $N_2O$ production during drought, leading to small $N_2O$ budgets in dry summer (Borken and Matzner, 2009; Congreves et al., 2018; Harry et al., 2021).

Autumn and spring $N_2O$ fluxes varied relatively little between years with different weather conditions,

indicating weaker sensitivity of spring and autumn $N_2O$ fluxes to seasonal weather conditions. $N_2O$ fluxes during autumn have been low also in part of the previous studies (Maljanen et al., 2003; Luo et al., 2012), but Pihlatie et al. (2007) and Alm et al. (1999) found increased autumn $N_2O$ fluxes after litter fall in drained peatland forest sites. Site-specific differences could alter contributions of different seasons to annual $N_2O$ budgets and affect sensitivity of $N_2O$ budgets to differing conditions in different seasons.





As the climate changes, the typical weather conditions for each season are predicted to change. In northern latitudes, winters are expected to become warmer and wetter, and summer droughts are expected to become more frequent (Zhao and Dai, 2017; IPCC, 2021). The high year-to-year variability in $N_2O$ fluxes, which was largely attributed to variation in summer and winter weather conditions, may imply changes and increased variability in annual $N_2O$ budgets if weather patterns of these seasons change and the frequency of extreme weather events
increases due to climate change.

### 4.2. Linkages to spatial variation

   Capturing temporal patterns of $N_2O$ fluxes from six chambers allowed us to explore the linkages between the spatial and temporal patterns of $N_2O$ fluxes across different measurement years. Lower $N_2O$ fluxes from three of
the chambers compared to high $N_2O$ fluxes measured in the other three chambers demonstrate the often spatially variable nature of $N_2O$ even on a small scale within a few tens of meters (Groffman et al., 2009; Hénault et al., 2012; Jungkunst et al., 2012).

   What was notable was that the spatial differences in $N_2O$ fluxes between chambers were persistent across different years. The mean and the maximum daily mean fluxes were consistently larger for the high-flux chambers
(Chambers 1–3), although differences between chambers were smaller during the low-flux year 2018 due to a larger decrease in $N_2O$ fluxes in high-flux chambers compared to low-flux chambers. Despite large temporal variations in flux within and between years, the spatial patterns of $N_2O$ flux remained throughout the measurement period.

   The persistence of spatial variation implies that spatial variation of $N_2O$ flux is controlled by long-term controls that persist throughout years with different weather conditions. The long-term controls could include, for
example, spatial variation in soil properties (e.g. pH, porosity, C and N content) or placement of plant roots that have both been suggested to affect the spatial variation of $N_2O$ fluxes even on a very small scale within the soil (Butterbach-Bahl et al., 2002; Jungkunst et al., 2012; Kuzyakov and Blagodatskaya, 2015). However, it must be noted that results regarding the causes of within-site spatial variation have been highly variable between different studies, and few studies have managed to explain spatial variation well (Ball et al., 2000; Butterbach-Bahl et al.,
2002; Yanai et al., 2003; Giles et al., 2012; Jungkunst et al., 2012). The linkages between soil properties, vegetation and $N_2O$ fluxes are complex, with interactions making the relations between the $N_2O$ flux and soil system difficult to understand.

   In the present study, the high-flux chambers had fewer trees near them than low-flux chambers, and the distance to nearby trees was shorter in low-flux chambers (Fig. 1, Table S1). This could indicate the importance of
trees shaping the spatial patterns of peatland forest floor $N_2O$ fluxes, similar as suggested by Butterbach-Bahl et al. (2002) in mineral soil forest. Trees may have impacted the availability of nitrogen through nitrogen uptake and nitrogen inputs to soil above and below ground (Kaiser et al., 2011; Kuzyakov and Blagodatskaya, 2015; Hu et al., 2016). Since trees also affect the forest microclimate and soil conditions by shading and affecting transpiration and distribution of rain fall in forest (Butterbach-Bahl et al., 2002; Aalto et al., 2022), variation in tree cover could have
contributed to the spatio-temporal dynamics of peatland forest $N_2O$ fluxes.



Although the chambers had persistently different levels of the $N_2O$ flux throughout the study period, the chambers had clear similarities in the temporal dynamics of the $N_2O$ flux (Fig. 4). High-flux and baseline flux periods identified for each chamber occurred often at the same time. $N_2O$ flux time series, especially among high-flux and small-flux chamber groups, also correlated, implying shared temporal patterns but stronger similarities between chambers with a more similar flux level. Similarities in the temporal flux patterns between the chambers indicate that the changes in the soil environmental conditions affect $N_2O$ fluxes relatively similarly despite the large spatial variation in flux.

Previous studies using manual chambers in agricultural settings with more variable soil conditions have found partly opposing results. Temporal patterns have been either variable or shared across space (Velthof et al., 2000; Krichels et al., 2019), with both findings mainly attributed to the spatio-temporal variation of soil moisture conditions. Soil moisture data from the individual chambers were not available here, but the chambers seemingly reached soil conditions triggering $N_2O$ production at similar times, although the resulting $N_2O$ flux level varied between chambers. In the presence of more topographical variation as in the study by Krichels et al. (2019), spatial variation in soil conditions could have led to more variable temporal patterns in $N_2O$ flux across space if triggering conditions of $N_2O$ production were reached at different times in different parts of the area. In the present study area, the factors causing the high and temporally persistent spatial variation in flux have not affected the way fluxes respond to temporal variation in soil conditions leading to similarities in temporal dynamics of the flux.

Differences in the temporal patterns between the high-flux and low-flux chambers were mainly related to the length and relative height of the high-flux periods as well as to the exact timing of the peak top within the high-flux periods. This has likely decreased the correlation of the temporal flux patterns between high-flux chambers and low-flux chambers. Since soil temperature variables were able to explain differences in temporal patterns of N2O flux between most chamber pairs, $N_2O$ peak length, timing, and relative height of flux peaks could be further shaped by spatial differences in the magnitude by which $N_2O$ fluxes respond to temperature conditions.

### 4.3. Freeze-thaw cycles

Increased winter $N_2O$ fluxes occurred in different phases of freeze-thaw cycles; during the onset of freezing periods, during repeated freeze-thaw events in the middle of the winter and during or after thawing in late winter and spring. Increased $N_2O$ fluxes in different phases of freeze-thaw periods can be seen, for example, in winter 2016–2017 (Fig. 6), with increased $N_2O$ fluxes measured during the early winter freezing as well as during and after part of the short freezing periods later in winter and spring.

Previous results about the timing of increased $N_2O$ fluxes regarding freeze-thaw cycles have been variable. Some studies report increased $N_2O$ fluxes during the freezing period (Papen and Butterbach-Bahl, 1999; Teepe et al., 2001; Maljanen et al., 2009, 2010; Ruan and Roberston, 2017), while part of the studies report high fluxes mainly during and after the thawing (Koponen and Martikainen, 2004; Pihlatie et al., 2010; Luo et al., 2012; Molodovskaya et al., 2012). Although here, the spring thaw resulted in a steady increase in $N_2O$ flux, with peak $N_2O$ flux reached later in spring or early summer, short-term $N_2O$ flux peak during soil melting was not observed. High




variability in the temporal patterns of winter and spring N$_2$O flux in different studies highlights the need to understand the causes of site-specific differences that create variable winter N$_2$O flux patterns.

The highest winter N$_2$O fluxes typically occurred in the early winter soon after the soil freezing at the time when frost reached 5 cm depth (Fig. 6 and 7, S5). Winter high-flux periods with peak N$_2$O fluxes clearly elevated from the baseline flux level were generally only measured during winters when soil frost reached 5 cm depth several times (winters 2015–2016 and 2016–2017) and only during freeze-thaw cycles occurring at 5 cm depth (late winter 2016–2017). The importance of deeper soil freezing rather than freezing only of the soil surface may indicate that the winter N$_2$O fluxes during freezing may have originated from the freezing peat rather than from the freezing litter

at the surface of the soil. The importance of the severity of ground frost and frost depth affecting N$_2$O fluxes has also been suggested by others (Nielsen et al., 2001; Koponen and Martikainen, 2004; Luo et al., 2012). The conclusion about the possible source of winter N$_2$O fluxes in the topsoil peat rather than in the soil surface litter differs from the results of Pihlatie et al. (2010) in a nutrient-poor peatland forest site. More nutrient-rich peat with a low C:N ratio may have favored N$_2$O production in peat (Regina et al., 1998; Ojanen et al., 2010). Higher nitrogen

availability in peat may have enabled a stronger link between winter N$_2$O fluxes and conditions experienced in the peat. Site-specific differences in nutrient availability in different parts of the soil may affect the sensitivity of winter N$_2$O fluxes to frost depth.

### 4.4 Delayed responses and interactions

The results of this study indicate general importance of lagged soil moisture and WTL conditions affecting N$_2$O fluxes on the short time scale of 1–7 days. Peak N$_2$O fluxes were reached sometimes several days after the highest surface soil moisture and WTL values were measured (Fig. 9, S6). Studies mostly from mineral soils have found no lags or lags of a few hours between the soil moisture peak and the highest N$_2$O fluxes, while others have found lags of a maximum of two days (Firestone and Tiedje, 1979; Smith and Tiedje, 1979; Song et al., 2022). In

this study, the lag time between surface soil moisture peak and the peak N$_2$O fluxes was typically at least two days, with indications for longer lags than seven days in some chambers. Long delays between the soil moisture peak and peak N$_2$O fluxes may be due to the ability of peat to retain moisture and therefore retain anaerobic microsites in soil for a longer time compared to most mineral soils (Päivänen, 1973; Walczak et al., 2002).

     Differences in the importance of different variables and their lags between chambers may indicate varying

lag-times and sensitivities to different soil moisture and WTL conditions across space. Despite spatial differences in lag times and differences in the most important variables for which the lags were identified, the highest N$_2$O fluxes on unfrozen soil were reached on intermediate soil moistures (0.3–0.4 m$^{-3}$ m$^{-3}$) after the soil had started to drain and WTL started to decrease after a precipitation event. The optimal conditions for high N$_2$O fluxes on intermediate soil moistures could be explained by the simultaneous occurrence of oxic and anoxic soil microsites that allow

simultaneous nitrification and denitrification in draining soil (Bateman and Baggs, 2005; Wang et al., 2021; Song et al., 2022).

     Although models were not run to different seasons separately, the response of N$_2$O fluxes to soil moisture peaks was slower in autumn. Lag times between peak N$_2$O fluxes and soil moisture peak increased and the height of



the $N_2O$ flux peaks decreased from summer towards late autumn. Lower temperatures in autumn leading to

decreased microbial activity and decreasing availability of N from decomposing peat in colder soil may explain lower fluxes and slow response of $N_2O$ fluxes to soil moisture peaks in autumn (Holtan-Hartwig et al., 2002). The finding reminds us of the importance of interactions affecting seasonal patterns of $N_2O$ fluxes.

## 5. Conclusions

The study shows extremely high temporal and spatial variability in peatland forest $N_2O$ fluxes with persistent spatial patterns and common temporal dynamics across space. The considerable small-scale spatial variation in $N_2O$ fluxes was persistent in time and is therefore likely to be influenced by relatively long-term controls in the soil. The temporal variation of $N_2O$ flux was instead strongly influenced by seasonal weather conditions such as precipitation, snow depth and drought. Temporally varying soil environmental conditions affect

$N_2O$ fluxes through complex responses and interactions, leading to high temporal variation in $N_2O$ flux between years as well as within and between seasons. Responses of $N_2O$ fluxes to environmental conditions include time lags that further shape temporal patterns of $N_2O$ fluxes.

The observed high peatland forest $N_2O$ emissions highlight the role of $N_2O$ emissions originating from non-agricultural systems and the importance of considering the spatio-temporal dynamics of highly seasonally variable

$N_2O$ fluxes, especially in boreal regions with strong seasonal patterns. The results indicate high importance of summer precipitation and winter temperature and snow conditions for seasonal and annual $N_2O$ budgets, and thus the possibility of increased annual variability in $N_2O$ emissions as seasonal weather conditions change in a warming climate.

**6. Appendices**

**Appendix A. Thermal seasons**

Thermal winter was the season with daily mean air temperatures persistently below 0°C and thermal summer a season with daily mean air temperatures persistently above 10°C (Ruosteenoja et al., 2011; Finnish Meteorological Institute, 2023). During spring and autumn, temperatures varied between 0–10 °C. Cumulative

temperature sums of daily mean temperatures were then used to identify starting days of the thermal seasons at which temperature goes persistently above or below the seasonal temperature threshold. The starting day of the thermal winter was the day after the annual cumulative temperature sum reached the maximum. The starting day of the thermal spring was the day after the minimum cumulative temperature sum was reached. Starting days of thermal summer and autumn were calculated similarly but by extracting 10 °C from the air temperatures before

calculating the cumulative temperature sum (modified temperature sum). The day after the minimum modified temperature sum was reached was defined as the starting date of the summer, while the maximum modified cumulative temperature pointed the onset of thermal autumn.




**Appendix B. Evaluating the model performance**

R$_2$ of the chamber-specific models used in the analyses varied between 0.72 and 0.85 in OOB data, and between 0.60 and 0.69 in training period evaluation data (30 % of training period data) (Table B1). When predicting N$_2$O fluxes outside the training period (fourth measurement year), R$_2$ varied between 0.02 and 0.69. Performance of N$_2$O gap-filling models was tested only using OOB data and evaluation data within the whole measurement period (30 % of data). For gap-filling models, R$_2$ in OOB data varied between 0.71 and 0.84, while R$_2$ in evaluation data

varied between 0.67 and 0.78 (Table B2).

For the models used in the analysis, the poor prediction accuracy outside of training period, especially in chambers 3, 4, and 6, was likely due to overestimation of the general flux level during the relatively dry year 2019, which was excluded from the training period (Fig. S8).The model was also unable to predict anomalous high-flux period in low-flux winter 2018–2019 in Chamber 5 likely due to a lack of chamber-specific soil temperature data

deeper in the soil. The temporal patterns of the flux otherwise followed temporal patterns of measured fluxes relatively well. Poor prediction accuracy outside the training period in part of the chambers indicates that predicting N$_2$O fluxes to a year with distinct environmental conditions compared to the years in the training data may lead to large under or overestimation of N$_2$O fluxes. The used models could benefit from additional explanatory variables, such as redox potential or availability of different forms of nitrogen (Rubol et al., 2012; Saha et al., 2020). Including

additional soil variables in the model could decrease the need to have excessively large model training periods to accurately predict and gap-fill N$_2$O fluxes.

**Table B1: Model performance in evaluation datasets. Out of bag (OOB) data refers to data left outside model training in Random forest with conditional inference trees, evaluation data within training period refers to 30 % of data randomly left aside for model evaluation and evaluation data outside training period refers to the fourth measurement year outside model training period (3 years).**

| Chamber | Evaluation data | RMSE | R$^2$ |
|---|---|---|---|
| 1 | OOB | 138.8 | 0.75 |
| | Within training period | 134.9 | 0.60 |
| | Outside training period | 113.7 | 0.67 |
| 2 | OOB | 105.7 | 0.84 |
| | Within training period | 106.0 | 0.69 |
| | Outside training period | 85.1 | 0.69 |
| 3 | OOB | 81.0 | 0.72 |
| | Within training period | 93.7 | 0.64 |
| | Outside training period | 75.7 | 0.02 |
| 4 | OOB | 36.3 | 0.83 |
| | Within training period | 29.5 | 0.77 |
| | Outside training period | 56.6 | 0.01 |



| | | | |
|---|---|---|---|
| 5 | OOB | 14.5 | 0.85 |
| | Within training period | 12.7 | 0.65 |
| | Outside training period | 22.0 | 0.33 |
| 6 | OOB | 10.2 | 0.85 |
| | Within training period | 10.3 | 0.68 |
| | Outside training period | 17.0 | 0.03 |

**Table B2: Performance of gap-filling models on evaluation datasets. Out of bag (OOB) data refers to data left outside model training in Random forest with conditional inference trees and evaluation data within training period refers to 30 % of training period data that was randomly left aside for model evaluation. Training period of gap-filling models covers the total measurement period (4.5 years).**

| Chamber | Evaluation data | RMSE | $R^2$ |
|---|---|---|---|
| 1 | OOB | 118.3 | 0.80 |
| | Within training period | 124.7 | 0.67 |
| 2 | OOB | 90.2 | 0.84 |
| | Within training period | 86.6 | 0.78 |
| 3 | OOB | 80.7 | 0.74 |
| | Within training period | 62.1 | 0.69 |
| 4 | OOB | 30.3 | 0.83 |
| | Within training period | 28.6 | 0.76 |
| 5 | OOB | 16.7 | 0.71 |
| | Within training period | 14.0 | 0.71 |
| 6 | OOB | 9.9 | 0.82 |
| | Within training period | 9.7 | 0.72 |

## 7. Data availability

Flux data and supporting environmental data are available at: https://doi.org/10.5281/zenodo.8142188 (Rautakoski et al., 2023a). R codes used in data analysis are available from the corresponding author by request. Python codes used in flux calculation and R codes used in data analysis are available from the corresponding author by request.

## 8. Supplement

The supplement of the article is available at: https://doi.org/10.5281/zenodo.8141569 (Rautakoski et al., 2023b).

## 9. Author Contributions

AL, MA, MK and PO set up the study design. Field maintenance of measurement systems was carried out by MK, AL and PO. Fluxes were calculated by MK and filtered by HR. Data analysis and modelling was carried out by HR with the support of AL and JM. HR wrote the article with the help of co-authors.



## 10. Competing interests

The authors declare that they have no conflict of interest.

## 11. Acknowledgments

We thank the Academy of Finland (Biogeochemical and biophysical feedbacks from forest harvesting to climate change – BiBiFe, Grant no. 324259; Managing Forests for Climate Change Mitigation – FORCLIMATE, Grant no.
347794) and the Maj and Tor Nessling foundation (Grant no. 201700450) for funding the work. We also thank for the support by the ACCC Flagship funded by the Academy of Finland (Grant no. 337552) and the Ministry of Transport and Communications through the Integrated Carbon Observation System (ICOS) and ICOS Finland. DeepL Write (DeepL SE, 2023) was used to improve the language of the article.

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
