# Peer review of "Exploring temporal and spatial variation of nitrous oxide flux using several years of peatland forest automatic chamber data"

_EGUsphere, 2023_

## Author Response (AR1)

**Information for the reviewers and the Editor:**

Line numbers at the beginning of each reviewer comment refer to line numbers of the previously reviewed version of the manuscript. These line numbers have been placed inside parentheses. Line numbers in our responses refer to line numbers of the revised manuscript. In the track-changes-version of the manuscript, the ==meaningful changes, such as sentences that have been added or modified based on reviewer comments, have been highlighted in yellow==. These major changes have also been explained in comments shown on the right-hand side panel. Minor changes, such as improvements in language, grammatical changes and changes resulting from the restructuring of the paragraphs, are marked by the Word track-changes function.  shows the old or removed version of the text and blue text shows new modified version of the text. In this file, our responses to the reviewers are numbered from 1 to 50.

**Relevant changes in the manuscript:**

- The Materials and methods section has been expanded to include more information on the automated chamber system, data processing, and machine learning. Materials and methods section now better supports the interpretation of results related to variable importance (VI) and accumulated local effects (ALE).
- The Results section has been made more compact and easier to understand by removing minor details and clarifying the content.
- Discussion has been made more compact and restructured to better highlight important findings. Results are now discussed more in the context of climate change.
- Results on soil moisture are supported by results on WTL and precipitation.

**Reviewer 1:**

General response:

We thank you for all your comments and suggestions regarding the article. The results section and discussion are now more compact and parts that were difficult to understand are now clearer. In particular, we have focused on how we present and discuss the results related to different years and chambers, as the amount of specific results related to a particular year or chamber was sometimes larger than necessary. We hope that these changes will help the reader better understand the content and that minor details won't distract from the main points of the article.

Specific comments:

Reviewer (Lines 158-159): Can you add in when measurements began for each variable? I had assumed that all environmental variable measurements began at the same time as the N2O flux measurements (although the start date for those is also not mentioned), but the next paragraph about water table level suggests otherwise.

1. Response: The measurement period times for each environmental variable were added. More exact start and end times of the chamber measurement period were added as well. We also more clearly state how the lack of soil moisture measurements at the end of study period was considered in the analysis.

See lines:
180-181: *"The measurements of air and soil temperatures were ongoing throughout the study period, but the soil moisture measurements ended half a year earlier than automatic chamber measurements (April, 2019)."*
192: *"Precipitation was measured at the site throughout the study period"*
253-254: *"The fourth year of measurements until soil moisture measurements ended (June 2018 – April 2019) was left aside for evaluation…"*

Reviewer (Lines 171-172): Can you add in what variables were used to model WTL before WTL measurements began in 2015?

2. Response: We apologize for being unclear. We added a clearer explanation about the modelling of WTL, including the explanatory variables used in the model and short description about the modelling process.

   See lines:

   187-191: *"WTL data from Chambers 1–2, seven other WTL loggers at the study site and precipitation were used as explanatory variables in the gap-filling model. Modeling was done first for the logger with the least amount of missing data, after which the gap-filled WTL time series was added to the model as an explanatory variable to increase the predictive power of the model for the variables with more missing data (evaluation data R2 = 0.90–0.97)."*

Reviewer (Lines 175-180): How close were the nearest weather stations to the study site? Can you please add in that detail?

3. Response: We added the distance to the nearest weather station (35 km) in the methods section.

   See line 104: *"…automatic weather station (Jokioinen Ilmala, 1991–2020, 35 km from the study site)"*

Reviewer (Lines 477): I know that most automated N2O flux datasets are restricted to small areas within an ecosystem due to how far chambers can extend away from a single gas analyzer. Nevertheless, I am not comfortable with characterizing "N2O budgets of the study site" based on a small area that may not be representative of the entire study site. Perhaps this can be more carefully worded (e.g., "study area" instead of "study site") to avoid the reader assuming inference beyond what is valid.

4. Response: This is a good point since the automatic chambers indeed cover only a small area and the word "study area" can be understood as a much larger area around the chambers. More careful wording referring to the area the N2O budgets represent, ("chamber area" or "study area"), is now used in the revised manuscript. The possibility of automatic chamber N2O budgets not representing the whole site is also mentioned in the discussion.

   See lines:
   450-453: *"The N2O budgets of our six automatic chambers are unlikely able to represent the N2O budget of the whole site, but the mean annual N2O budget of the chamber area greater than 950 mg N2O m-2 y-1 in the two full study years out of three underlines the role of drained nutrient-rich peatland forest as hotspots for N2O emissions"*
   454: *"…low C:N ratio likely explains the high N2O budgets of the chamber area."*

Reviewer (Lines 490-498): Here a contrast is drawn between the relatively less important spring N2O fluxes at this study compared to other boreal and temperate studies in which spring N2O fluxes have accounted for a greater proportion of the annual N2O budget. Rather than simply pointing out this difference, it would be more meaningful to explore what could have caused this difference because it then allows readers to hypothesize whether spring N2O fluxes might be important in their study systems. What might be different about environmental conditions during spring at this site compared to the other studies? Or what might be different about soil properties at this site (perhaps in combination with site differences in weather) that would make mechanisms for high spring N2O fluxes not in play here?

5.  Response: The possible explanation for the small spring fluxes is now discussed more broadly. The results suggest that the main control of spring N2O emissions at the study site is soil temperature which might indicate that small spring fluxes might be at least partly due to substrate limitation after winter.

    See lines 496-502: "*The strong temperature dependence of spring N2O fluxes may indicate that the substrate for the spring N2O production comes from the decomposing peat and litter in the warming soil. Temperature dependence of spring fluxes could be related to drained peat soil, where the major source of N is known to be decomposing peat (Martikainen et al., 1993). Different responses to thawing in winter compared to spring might be related to decreasing availability of N from early winter towards spring (Koponen and Martikainen, 2004; Congreves et al., 2018), which also likely explains the tendency for stronger response to freeze-thaw cycles in early winter.*"

Reviewer (Line 616-618): I do not understand the logic behind this explanation of long lags between peak soil moisture and peak N2O fluxes. If peat retains moisture and therefore anaerobic microsites, then it seems that this would allow high denitrification rates to occur long after a rainfall event or after the water table level drops. Could you clarify the reasoning that peat retaining moisture would lead to a delay in N2O fluxes responding to high soil moisture?

6.  Response: We apologize for being unclear. We have now added a clearer explanation about the causes of soil moisture-related lags in the revised manuscript. However, we admit that the specific conclusion made here related to soil moisture can be weak due to the lack of chamber-specific soil moisture data. We will also indicate this more clearly in the revised manuscript and support findings with WTL and precipitation.

    See lines 568-577: "*Based on this study and previous laboratory studies, we suggest that after a period of high N2O reduction activity and therefore relatively low N2O fluxes from denitrification in the wet soil soon after rain, N2O production in the draining soil increased (Firestone and Tiedje, 1979; Russow et al., 2000; Congreves et al., 2018). As soil continued to drain, conditions for simultaneous nitrification and denitrification became optimal (Bateman and Baggs, 2005; Wang et al., 2021; Song et al., 2022), further increasing N2O production and leading to the peak N2O flux some days after rain. The ability of peat to retain moisture could extend the time for soil drainage after rainfall and thus time before optimal conditions for N2O production are reached. Hydrophobic properties of dry peat soils can also extend the time before N2O fluxes respond to soil wetting (Borken and Matzner, 2009) contributing to longer lag times. To determine exact lag times in response to soil moisture peaks, chamber-specific soil moisture data would be required.*"

Reviewer (Lines 643-645): Again, I caution against using data from a 20 m x 20 m area of the study site to characterize boreal peatland forest N2O emissions or non-agricultural N2O emissions in general. Given the high spatial variation within this small area, can we really assume that the results can be extrapolated across larger spatial extents?

7.  Response: We agree that the fluxes of the small study area do not necessarily represent fluxes on larger scale. We believe that information about the processes and their controls, however, can contribute to the understanding about spatio-temporal variation of N2O even on relatively large scale. The conclusions section has been modified to avoid making such a direct link between emissions from our study site and other boreal peatlands.

    See lines 597-598: *"The observed high N2O fluxes from the peatland forest highlight the role of nutrient-rich drained peat soils as hotspots for N2O emissions in the boreal region"*

Technical comments:

Reviewer (Line 170): Should this state that chambers 4-5 shared at WTL sensor? Based on Figure 1, chambers 3-4 are on opposite ends of the sampling area so it seems unlikely that they would share a WTL sensor.

8.  Response: We apologize for the mistake in the chamber number. The chamber number was corrected.

    See line 184: *"Chambers 1–2 and 4–5 shared a WTL logger…"*

Reviewer (Lines 506): Missing a comma after "especially during summers"

9.  Response: Comma was added and sentence modified.

    See line 503: *"High-flux periods during the growing season, especially in the summer, were associated with precipitation events that increased soil moisture and raised WTL"*

**Reviewer 2:**

General comments:

The authors present a substantial dataset on N2O fluxes from a drained peatland forest in Finland. The manuscript follows a clear structure and is understandable, even though the writing could be improved with the usage of commata and/or a native English speaker to look through (some specific comments below). The material and methods section could also use a bit more detail here and there to understand the experimental design and data processing without having to read several other papers first. The data analysis is state of the art, insightful and well chosen for the purpose of the study's aim. The figures and tables are very nicely done and helpful. The results could be a bit more focused and shortened.

My main question is concerning the data resolution on which the analysis was performed on. The study wants to highlight the variability of environmental driver on a spatial and temporal scale, yet the analysis was done on daily flux averages. N2O has a very intense and short-lived nature and can vary significantly on an hourly basis or less. Why was the analysis not performed on the hourly

measurements, since you have such a great resolution in front of you? I do understand that the computing time would increase significantly. So maybe start with shorter timeframes before, during and/or after high-flux periods. This could give valuable new insight, additional to what is already established in the literature and highlighted here again, that freeze-thaw and rewetting events lead to high N2O emission periods. This dataset is very valuable to publish because of the length and temporal resolution but I think there could be more in it than currently analysed and discussed.

On the other hand, the conclusion of lagged N2O flux responses to soil wetting with e.g. VI scores is a bit weak with the soil moisture measurements at hand, which were not located at the chambers but at a weather station 75m away and only once without any replicates. From the literature we know that soil moisture is one of the most important variables but also varies significantly on the micro-scale. The conclusion on soil moisture is a bit contradicting in the text as well. The random forest model identified that lagged soil moisture was important but not as much as the unlagged soil moisture (L26). The VI score is clearly higher for the unlagged soil moisture (L405). But the ALE found lagged soil moisture more predictive for high fluxes (only when the soil moisture was high, L425). I believe that overall, this central statement around lag time after soil wetting developed from the analysis of soil moisture here, is not fully supported by the experimental setup. Soil moisture measurements per chamber might have given a much clearer picture concerning any potential lag time but with the data as is, I would shift the focus towards more reliable analysis. Conclusions drawn from the soil moisture variable can be stated but should be considered very carefully.

With the suggested changes mentioned in the general, specific and technical comments, I would recommend publishing this manuscript in the EGUsphere.

10. **General response:** We thank you for the valuable feedback. The manuscript has now been revised according to your comments and suggestions. The Materials and methods section now includes more information about the automatic chamber system and other used methods (high-flux period identification, VIs, ALEs). The text in the Results section as well as the figure captions have been modified to help the reader interpret the results (density plot, VIs, ALEs). See responses number 11, 15, 16, 18, 19, 28 and 29). We shortened the Results section (- 450 words) section by focusing more clearly on the main results and by removing unimportant details or repetitive sentences from the text. Discussion was restructured to better highlight the important findings (-400 words). Topics that needed further discussion are now discussed more broadly (lagged responses, winter fluxes, climate change. See responses number 6, 34 and 35).

    **Concerns related to daily resolution:**
    The daily resolution was chosen for two reasons. Firstly, the temporal variation in N2O flux was dominated by weekly, monthly and seasonal variation instead of hourly variation. Hourly peaks that were not captured by daily means were not typical. When N2O peak started, the N2O fluxes steadily increased day by day. Secondly and most importantly, the automatic chamber system seemed to create an artificial diurnal cycle of N2O from which the natural diurnal cycle was impossible to separate from. We considered filtering out problematic hours but setting criteria to define problematic fluxes would have been very difficult and led us possibly to delete large amounts of good quality data as well. We used daily averages since they capture the main temporal variation seen in N2O flux (daily to yearly) and are less affected by the over or underestimation of fluxes caused by the artificial diurnal cycle. We admit that the artificial diurnal cycle creates an additional source of uncertainty to the reported annual N2O budgets,

and we have now added a mention about this in the article. As a response to your concerns, we have explained the choice of the temporal resolution better in the methods section and included a better description about turbulence problem and issues related to it.

See lines 148-160: *"Daily mean N2O fluxes from each chamber were used in the analysis because the automatic chamber system seemed to create an artificial diurnal cycle of N2O from which the possible natural diurnal cycle could not be separated. The artificial diurnal cycle was caused by the difference in turbulence between the ambient air and chamber headspace, as previously reported for CO2 and CH4 (Koskinen et al., 2014; Korkiakoski et al., 2017). During calm periods, especially during summer nights, the transfer of N2O from soil pores to the atmosphere slowed down, leading to increased N2O concentration in the soil. When the chamber closed and the turbulence increased because of the fan, the N2O from the soil pores was vented to the chamber headspace air, leading to overestimated flux. The opposite phenomenon probably occurred in windy conditions, resulting in underestimated flux. Based on our experience, automatic chamber fluxes measured in drained peatlands with dry and porous peat soil are particularly sensitive to this phenomenon (Koskinen et al., 2014; Korkiakoski et al., 2017). Hourly N2O flux peaks were not typical in the flux data, and daily mean N2O fluxes thus well represent the main characteristics of the temporal variation. It should be noted that the artificial diurnal cycle creates an additional source of uncertainty in the reported N2O budgets."*

**Concerns related to lagged soil moisture:**
The lack of chamber-specific or close-to chambers soil moisture data was an unfortunate limitation in our study. We have now been careful with the conclusions made from soil moisture results alone and supported findings with WTL and precipitation that better describe conditions at the chamber site. The results about delayed effects of soil moisture, WTL and precipitation are in line with each other, creating a coherent story about the importance of lags in response to soil wetting. Therefore, we decided to keep the delayed response to soil wetting as one of the important conclusions of the study, but we bring up more openly the uncertainties related to soil moisture both in the Material and methods section and in the Discussion. See response number 39.

Referring to the example you brought up, as the means of the lags for soil moisture show (lines 410-417), the lags are generally important and dynamics seen in the ALE plots for soil moisture, WTL and precipitation further underline the importance of lags after soil wetting. The fact that the unlagged soil moisture was often more important than individual lags doesn't make the conclusion about the general importance of lags any weaker. The model is likely able to predict the fluxes relatively well also without lags, just based on the general level of soil moisture. The importance of lag dynamics varies in time, since lags are only present in certain conditions and the lag-times vary between seasons. This decreases the overall VIs of lags despite them being important part of the dynamics related to N2O response to precipitation. See also response number 29.

Understanding machine learning models is often a challenge, although tools like variable importance scores or accumulated local effect (ALE) plots help with interpreting the results. Regarding VI scores we want to clarify one thing here and in the revised manuscript. Variables and their lagged versions are highly correlated, which is considered in model and method

selection (Strobl et al., 2007, Strobl et al., 2009, Apley and Zhu, 2020). Out of the set of correlating explanatory variables (unlagged + lagged), the one that gets the highest VI score out of all the correlating unlagged and lagged variables has the highest ability to predict N2O flux, for one reason or another. The reason can be something very data-specific (typical for many machine learning models) or explained by real phenomena (for example, the unlagged variable is truly more important than others). For this reason, the interpretation of the lag-specific VI scores (Fig. S6) should be done carefully. The mean importance of lags across chambers and total VIs (sum of unlagged and lagged VIs of a variable) better describes the overall importance of lags with less risk for very data-specific conclusions. For this reason, we replaced the lag-specific VI figure with a figure showing the total VIs of each variable (Fig. 8). Lag-specific VIs are shown in the supplements. ALE plots that show the dynamics between lagged environmental conditions and model predictions visualize the dynamics through which the lags affect the temporal patterns of N2O. Therefore, ALE plots are an important tool to understand the lag-dynamics without having to assume that lags should also be considered important explaining all the temporal patterns of N2O.

We are sorry for not making this clear in the first version of the manuscript. Besides changing the figure to more easily interpretable figure, we paid attention to the figure captions of ALE and VI figures and explained the interpretation of VIs and ALEs also in the Material and methods section.

See lines:
258-269: *"VIs and accumulated local effects (ALE) were used to interpret the modeling results. For easier comparison of VIs across chambers, the VIs of each chamber were scaled from zero to one (0 = least important variable, 1 = most important variable) and the total VIs of each variable were calculated (total VI = VI of unlagged variable + VIs of lags). The ALE method by Apley and Zhu (2020) was used to visualize the response of N2O flux to environmental conditions and their lags in the models. In ALE figures, ALE value (y-axis) zero refers to the mean predicted N2O flux, with a positive ALE value meaning larger and a negative value lower predicted N2O flux in a specific environmental condition (x-axis). ALE values for lagged environmental variables indicate the response of predicted N2O flux to previous environmental conditions. From the unlagged and lagged versions of each environmental variable, the one that received the highest ALE value for a given environmental condition was considered to represent the typical response time of N2O flux to that condition. In this article, the response time, or lag length in the presence of at least a one-day lag, refers to the time it takes for N2O to reach peak flux after the onset of a given environmental condition."*
404-408: *"Figure 8: Total variable importance (VI) of different environmental variables in explaining the temporal variation of N2O flux in random forest with conditional inference trees. Total VI is the sum of VIs of unlagged and lagged (1–7 days) versions of the variable. Rows in the matrix plot show VIs for different chambers and the mean VIs across all the chambers (Mean). VI scores are scaled between 0 and 1 (0 = no importance, 1 = highest importance) per chamber. Lag-specific VIs are shown in Fig. S6."*

Specific comments:

Reviewer (L139-140): Could you give more background information on the measurement procedure, such as flush time and/or time given to get the air circulated to reach a chamber-analyzer equilibrium? It must have taken some time for these big chambers. Is there an internal or external multiplexer involved to switch between chambers? Was there any ambient air measured? Was there any issue with drift (I do not know if these analyzers have overcome this due to the method)? What is the detection limit or threshold of the system (e.g. noise level of the system while sampling ambient air from an open chamber. When you are certain this is a small flux rather than noise? Did you adjust for temperature and air pressure fluctuations over the year for the flux calculations? How did you deal with snowy conditions? Did the chambers get buried in the snow or did you dig them out or lift them up with extensions? Etc.

**11.** Response: We added more detailed information about the automatic chamber measurement system, including minimum detectable flux, flow rate and information about the ambient concentration measurement. The drift of the analyzer was very minimal (maximum drift 0.1 ppb in 24 h), did not affect fluxes, and was therefore not considered in flux calculation. Effect on air temperature and pressure was considered in the flux calculation (See equation for flux calculation in Korkiakoski et al., 2017). The mean air temperature of each chamber closure and air pressure measured at the site was used and we added a mention about this in the manuscript. Information about the chamber maintenance in winter (extension collars, clearing snow, measuring snow depth of chambers) was also added.

See for example lines:
130-132: *"During winters, chambers were cleaned from snow and ice every 1–3 weeks and snow depth inside the chambers was measured to account for the effect of snow depth on chamber volume. During the winter 2016–2017, extension collars were used to better allow snow to fit inside the chambers."*
135-142: *"The analyzer had an accuracy of 0.01 ppb per second, corresponding to a minimum detectable flux (Nickerson, 2016) of 0.06 µg N2O m⁻² h⁻¹ in our chamber system. During each chamber closure, air from the closed chamber was pumped into the analyzer and back to the chamber headspace through plastic tubes (length 15 m, flow about 1 l/min). After each chamber closure, the airflow was switched to the next chamber. Ambient air was measured for at least 1 min between the chamber closures to allow concentrations in the tubes to stabilize back to the ambient level. Concentration data from the first 30 s of each chamber closure were not used in flux calculation to avoid possible pressure disturbance caused by the closing chamber affecting the flux (Pavelka et al., 2018)."*
145-146: *"Mean headspace temperature of the closure and air pressure measured at the site were used in the flux calculation."*

Reviewer (L146-148): This information cannot be understood easily without reading the references, which interrupts the reading flow. Please provide a bit more explanation here on how a fan can create a diurnal flux dynamic.

**12.** Response: We added a clearer explanation about the artificial diurnal cycle in the Material and methods section and also mentioned uncertainties related to it.

See lines:
150-158: *"The artificial diurnal cycle was caused by the difference in turbulence between the ambient air and chamber headspace, as previously reported for CO2 and CH4 (Koskinen et al.,*

*2014; Korkiakoski et al., 2017). During calm periods, especially during summer nights, the transfer of N2O from soil pores to the atmosphere slowed down, leading to increased N2O concentration in the soil. When the chamber closed and the turbulence increased because of the fan, the N2O from the soil pores was vented to the chamber headspace air, leading to overestimated flux. The opposite phenomenon probably occurred in windy conditions, resulting in underestimated flux. Based on our experience, automatic chamber fluxes measured in drained peatlands with dry and porous peat soil are particularly sensitive to this phenomenon (Koskinen et al., 2014; Korkiakoski et al., 2017)."*

159-160: *"It should be noted that the artificial diurnal cycle creates an additional source of uncertainty in the reported N2O budgets."*

Reviewer (L150): Could you add the analyzer position into the Figure 1b plot?

**13.** Response: Measurement cabin, where the analyzer is, is now added in the Figure 1b.

See also line 133-135: *"N2O concentration of the chamber headspace air was measured using a continuous-wave quantum cascade laser absorption spectrometer (LGR-CW-QCL N2O/CO-23d, Los Gatos Research Inc., Mountain View, CA, USA) that was placed in the measurement cabin close to the chambers (Fig. 1b)."*

Reviewer (L167-173): I would have thought that soil moisture varies much more between chambers rather than WTL. Did you see statistically significant differences between the WTL measurement locations?

**14.** Response: Water table level can be relatively variable in drained peatland forests due to for example variable distance to closest ditches and varying amount of transpiring trees. The differences in WTL between all the loggers were statistically significant, with chambers 3 and 6 having the highest WTL and chambers 4-5 the lowest WTL. These differences in WTL did not explain spatial differences in N2O flux.

See lines 336-337: *"Differences in WTL between chambers were statistically significant but were not associated with the spatial variation of N2O flux."*

Reviewer (L194-206): This concept of identifying high-flux periods is a not quite clear to me in the way it is described here and why you want to do this numerically. N2O fluxes can be incredibly short lived and do not always continue over several days or even hours for that matter. Is this categorization needed for your interpretation and correlation to seasonal dynamics? Could you state here why you need to identify these periods numerically on the daily resolution? (And please reconsider the order of sentences, this is very hard to follow. E.g. The second sentence of the paragraph is starting with 'finally', which implies that this statement is now finished but then the whole explanation on how you get to the 70% comes after it. I thing a few sentences could be deleted to make this shorter and clearer. E.g. 'High fluxes were measured less frequently (30%) compared to the more common low fluxes (70%), which was apparent from flux histograms (Fig. S2).' Then you can delete the next three sentences. And finish with the earlier mentioned sentence from above, adjusted to the context here: 'This histogram analysis concluded the 70% percentile threshold to define high-flux periods.')

**15.** Response: We thank you for the feedback regarding the explanation of high-flux period identification. We reformatted and clarified the explanation of high-flux period identification.

We wanted to identify high-flux periods to explore their length, timing and seasonality (Fig. 5). High-flux periods were also used to understand the linkages between environmental conditions and N2O peaks (Fig. 7) besides random forest. Identification of high-flux periods or characterization of the temporal variation could also be done entirely manually by looking at the timeseries, but this method is very sensitive to biases caused by our own thinking, especially in large datasets like ours. For these reasons, high-flux periods were determined numerically. Temporal variation in our data set is dominated by daily and seasonal variation instead of hourly variation, and very short-lived peaks are not typical. High-flux periods identified on daily resolution are enough to capture the main characteristics of temporal variation seen at the study site. We added a mention about this in the methods section.

See lines:
158-159: *"Hourly N2O flux peaks were not typical in the flux data, and daily mean N2O fluxes thus well represent the main characteristics of the temporal variation."*
208-212: *"To identify the high-flux periods, their length, seasonality and starting conditions, different thresholds to separate high-flux days from the baseline days were tested. High fluxes were measured less frequently compared to the more common low fluxes (Fig. S2). Any percentile threshold between 60–80 % separated high-flux days from the more common baseline fluxes relatively well, and the mean of these (70 %) was used. Days with the mean flux above the 70 % percentile were classified as high-flux days and the rest of the days as baseline days."*

Reviewer (L209-211): I must admit that I do not understand how and why the 'N2O flux of each chamber can be explained by flux of one other chamber' within a multiple linear regression. Could you maybe include some visual output in the supplemental material to makes this easier understood? And why do you want this? What does this correlation highlight?

**16.** Response: We admit that the explanation about the multiple linear regression test was not clear enough. Since multiple linear regression was very minor part of our study, we decided to drop it out.

Reviewer (L243): I have a general concern for the flux resolution you are using. You have such great data of 24 measurements per day and chamber. Why do you bring it to the daily resolution first before running the models to see which environmental parameters are driving the fluxes? You might get incredible insight from the short-lived emissions that happen besides freeze-thaw or dry-wetting events. Is there a time during the day where the fluxes are higher? Is, for example, in the morning of very dry days the dew a factor? If you want to figure out the environmental drivers, why exclude the diurnal dynamic in your data. You certainly have the resolution for it. Could you gap-fill the hourly rates? And the lag might be even less than a day worth too.

**17.** Response: We thank for the critical comment regarding the choice of temporal resolution. The daily mean N2O fluxes were used instead of hourly fluxes because the temporal variation of N2O flux was strongly characterized by variation from daily to yearly scale and automatic chamber system seemed to create artificial diurnal cycle in the flux especially during summer. Please see the response number 10 for more detailed explanation.

See lines:

148-150: *"Daily mean N2O fluxes from each chamber were used in the analysis because the automatic chamber system seemed to create an artificial diurnal cycle of N2O from which the possible natural diurnal cycle could not be separated.*
158-159: *"Hourly N2O flux peaks were not typical in the flux data, and daily mean N2O fluxes thus well represent the main characteristics of the temporal variation."*

Reviewer (L250): How did you analyze the prediction accuracy based on the R2 and RMSE? Are these parameters just calculated and stated or did you select the model based on the best R2 and lowest RMSE? Please a little more detail here.

18. Response: We did not perform model selection and a mention about this is now added in the article. R2 and RMSE were only used to evaluate model performance on OOB data, evaluation data within training period and outside training period. R2 and RMSE allowed us to evaluate model performance outside training data, which is important part of modelling. R2 and RMSE are commonly used metrics in model evaluation and therefore chosen. R2 was mainly used in the evaluation since the scale of the response variable does not affect it. We clarify this shortly in the revised manuscript.

See lines 255-256: *"The performance of the models on different evaluation datasets was analyzed using R squared (R2) and root mean squared error (RMSE). R2 was used to compare model performance between chambers. Variable selection was not done."*

Reviewer (L254-255): '…to visualize the response …, by illustrating how the predicted N2O flux values deviate from the mean predicted flux (ALE = 0) along the gradients of the measured environmental parameters' You describe it in the Figure 9 caption but not in the material and methods. Not everyone has the same knowledge than you. Please describe briefly or just copy from your caption.

19. Response: Thank you for the feedback. More information about ALE and its interpretation is now added in the methods section. The results related to ALE plots are also explained more clearly in the results section, including the mean identified lag-times also for WTL and precipitation.

See lines:
262-269: *"In ALE figures, ALE value (y-axis) zero refers to the mean predicted N2O flux, with a positive ALE value meaning larger and a negative value lower predicted N2O flux in a specific environmental condition (x-axis). ALE values for lagged environmental variables indicate the response of predicted N2O flux to previous environmental conditions. From the unlagged and lagged versions of each environmental variable, the one that received the highest ALE value for a given environmental condition was considered to represent the typical response time of N2O flux to that condition. In this article, the response time, or lag length in the presence of at least a one-day lag, refers to the time it takes for N2O to reach peak flux after the onset of a given environmental condition. The reported evaluation results (RMSE, R2), VIs, and ALE values are averages over 10 model runs."*
410-417: *"Accumulated local effects (ALE) for 7 cm soil moisture showed that the highest fluxes were predicted for the 1–7 days lagged soil moisture when the soil was moist (> 0.35 m-3 m-3). The lag with the highest predicted flux varied from 1 to 7 days between chambers with a mean lag of 4 days. Predicted fluxes were also high when soil moisture was low (< 0.1 m-3 m-3,*

*frozen soil). For WTL, the predicted flux was generally high when WTL was high (> –45 cm), with the highest predicted flux on average for 4 days lagged WTL. The predicted flux for unlagged WTL was low at high WTL, while the predicted flux for unlagged WTL increased with decreasing WTL. N2O flux was predicted to be the highest 4–7 days after precipitation with an average lag of 5 days between chambers when daily precipitation had been at least 5 mm."*

Reviewer (L269-270): Is the code somewhere published? And the data made available?

**20.** Response: Yes, data was made available and the simplified version of the code of the machine learning part of the study is now also published. You can find the data here (Rautakoski et al., 2023a and 2023b): https://doi.org/10.5281/zenodo.8142188 and https://doi.org/10.5281/zenodo.8141569. The code is available on Github: www.github.com/helenemilii/N2O_modeling. I'll be happy to share the full code for any interested readers.

See lines 652-655: *"Flux data and supporting environmental data are available at: https://doi.org/10.5281/zenodo.8142188 (Rautakoski et al., 2023a). Simplified R code of the machine learning part of the study is made freely available at: https://github.com/helenemilii/N2O_modeling. Python codes used in flux calculation and R codes used in data analysis are available from the corresponding author by request."*

Reviewer (L276-280): Here you describe the temperature variation and refer to Figure 2. Yet in Figure 2, you show anomalies only, nothing about this in the text.

**21.** Response: We changed the position of the figure reference and modified the text to better match the content of the figure.

See lines 290-293: *"The summers (June, July, August) 2015 and 2017 were colder (seasonal means 14.1 °C and 14.4 °C, respectively) than the long-term average (15.6 °C, Jokioinen-Ilmala 1991–2020), and winters (December, January, February) 2015–2016, 2016–2017 and 2018–2019 were warmer (seasonal means –3.4 °C, –3.0 °C and –3.5 °C, respectively) than the long-term average (–4.3 °C) (Fig. 2)."*

Reviewer (L288-290): This sentence is very confusing, consider rearranging. E.g. Soil moisture at 7cm and 20 cm were continuously lower in summer 2018 until 2019 (xx and yy respectively) compared to the entire measurement period mean (xx and yy respectively). Please consider this for the entire results, please always keep one statement with their numbers clearly separated and then compare them to something else. Next sentence as well, WTL was deeper in the summer and autumn 2015 (xx) than the mean of the study period (yy). I am getting lost in every sentence to which statement the value belongs to.

**22.** Response: We appreciate your comments related to the readability of the text. We have now made the text easier to read and values easier to interpret by moving some of the values that were inside parentheses to the actual text and by better indicating what the values inside parentheses are (words like "mean", "seasonal mean", "max." used inside parentheses).

See for example lines:
See lines 290-293: *"The summers (June, July, August) 2015 and 2017 were colder (seasonal means 14.1 °C and 14.4 °C, respectively) than the long-term average (15.6 °C, Jokioinen-Ilmala 1991–2020), and winters (December, January, February) 2015–2016, 2016–2017 and 2018–*

*2019 were warmer (seasonal means –3.4 °C, –3.0 °C and –3.5 °C, respectively) than the long-term average (–4.3 °C) (Fig. 2).*"

299-301: "*Soil moisture was continuously lower than the mean of the study period from the summer 2018 until the end of the study period (Fig. 3), with the study period mean being 0.28 m-3 m-3 for 7 cm soil moisture and 0.56 m-3 m-3 for 20 cm soil moisture.*"

Reviewer (L296): Again, I cannot follow here what you want to say about the snow cover periods, reaching the maximum in 2018-2019 and lowest in 2016-2017? What does this mean about the 'all winters had a period or periods of snow cover with the maximum measured… Or do you just mean 'All winters had a period with permanent snow cover. The maximum snow cover was measured…'

**23.** Response: Yes, you understood correctly. The first part of the sentence has now been removed and the sentence reformatted.

See lines 304-305: "*The snow cover was thickest in winter 2018–2019 (max. 52 cm) and thinnest in winter 2016–2017 (max. 11 cm).*"

Reviewer (L310): Figure 3: Very nice graph. The water table varies a fair bit spatially but does not really go closer to the surface than -20cm, while the 20cm soil moisture is not really picking up that dynamic. I would suggest including precipitation in this graph as well, to give more insight into the 7cm soil moisture dynamic, since there are no replicates for soil moisture, right? Overall I think this experimental setup (if continuing) would greatly benefit from more 7cm soil moisture replicates (i.e. per chamber), since N2O is highly correlated to the small-scale moisture dynamic!

**24.** Response: I strongly agree that having more soil moisture measurements and having them closer to chambers would be important. We were unlucky with the soil moisture sensors installed nearby the chambers, that were not working as they should have been. We have now added precipitation to Figure 3.

Reviewer (L334): I do not understand why flux time series correlations where done on chamber pairs. What is this information supposed to tell me? Their dynamic relates to one another? Is this something new?

**25.** Response: Temporal variation had clear similarities visually (see fig. 4) and correlation was one way to test similarity numerically. It was partly surprising since the spatial variation in N2O flux was so large between chambers. One could have assumed that places that have very different levels of N2O flux could respond to the variation in environmental conditions at least partly differently. There were differences between temporal variation but also strong similarities.

Reviewer (L340): Figure 4: Very nice. If you like, you could make the figure even more powerful by highlighting the high-flux periods in another color.

**26.** Response: High-flux periods have now been visualized with different color.

Reviewer (L390): Figure 6: Again, very nice and you could improve it by extending the 'soil surface <0 °C' shaded area to the other two plots in the graph, to make the N2O and moisture responses even easier to follow.

**27.** Response: We added the shading also to the soil moisture and WTL plot. We also added a precipitation panel.

Reviewer (L400): Figure 7: Not sure if I read this figure right, high-flux period density in different environmental conditions. How does the baseline period fit into this? Do you mean the density of high N2O fluxes, which are happening in the no-high-flux periods? Do you want to compare those? There is nothing in the text about it. Maybe add again to the caption that the scaling of 0-1 was done to compare the chambers. Not sure what I should take away from this graph. There is not much in the text other than referring to the temperature plot. Could you add an introducing sentence when bringing in figure 7 on what you want to compare/highlight here, with a take-away message that you want to discuss later?

**28.** Response: We modified the caption to help the reader with the interpretation of the figure. We have also made the link between the text and the figure stronger by discussing the more of the main results of the figure in the text. The aim of figure 7 is to visualize in which kind of condition the identified high-flux periods started and compare those to the conditions during baseline periods. The density is a way to visualize distribution. Histograms visualize the number of observations in each range of x-axis values, while density plots show the proportion of observations on different values of x-axis. In this case, the absolute proportions are scaled between 0 and 1 to simplify the interpretation of y-axis values between seasons and variables. The high-flux period density 1 points to the conditions when highest proportion of high-flux periods started in each season for each environmental variable. For example in spring, most high-flux periods started when soil surface temperature was close to 0°C, soil moisture at 7 cm about 0.2 $m^{-3}$ $m^{-3}$, soil moisture at 20 cm about 0.62 $m^{-3}$ $m^{-3}$ and WTL at about 55 cm depth. The idea with the baseline period densities is to show what kind of conditions there are in each season outside high-flux periods. If the conditions at high-flux period start are very distinct compared to the overall variation in conditions outside high-flux periods (See for example Fig 7b for summer), we could assume that only certain conditions trigger high-flux periods in that season. If the conditions triggering high-flux periods are the same as the conditions outside high-flux periods, there must be some other cause for the high-flux periods and the variable in question is not able to explain why high-flux periods start (See for example Fig 7d for winter).

See lines:
387-393: *"Figure 7: High-flux period starting conditions in each season compared to conditions outside the high-flux periods. Density plots show the distribution of high-flux periods starting on different a) soil surface temperature, (b) soil moisture at 7 cm depth, (c) soil moisture at 20 cm depth, (d) and water table level (WTL). The Y-axis shows scaled (0–1) proportion (%) of high-flux periods starting on conditions shown on the x-axis (1=most common high-flux periods starting condition, 0=no starting high-flux periods). For comparison, the variation in soil conditions during baseline periods is also shown (1=most common baseline period condition, 0=no such condition measured during baseline periods). All years and chambers are included."*
358: *"with most of the spring high-flux periods starting at soil surface temperatures 0–2 °C (Fig. 7)."*
360-361: *"Summer high-flux periods started after precipitation events at moist soil conditions (0.37–0.41 m-3 m-3) and during relatively high WTL (–35 to –50 cm depth) (Fig. 7)"*
364: *"Winter high-flux periods started on soil temperatures close to 0 °C (Fig. 7)"*

Reviewer (L 405): Could you please explain VI scores. Because without explanation, this whole part sound to me that the importance of the unlagged variables always scored higher than the lagged one, except precipitation. But in the abstract, you mention that the N2O responses to soil wetting with a lag period. Then it should be rain events rather then soil wetting, since the water table can contribute to wetting in 20 cm as well. Overall, I would be careful of the conclusions you get from the measurements of the weather station, since it is not representative for each chamber individually! Soil moisture can vary significantly on the small scale. This might explain why precipitation is behaving quite 'smooth' in your analysis. One would assume that all chambers received the same amount of rain. As sad as it is, I would consider dialing down the take-home message here, since I am not confident in the representation of soil moisture for the site's soil wetting overall.

29. **Response:** We added a paragraph about the interpretation of VIs in the Materials and methods section. The text related to the VI results is now also reformatted and made easier to interpret. We replaced the VI figure with a figure that visualizes total VIs (sum of unlagged and lagged) instead of individual VIs of all the lagged and unlagged variables. The new figure is easier to interpret and the risk for very data-specific conclusions is smaller (see explanation below). The more detailed VI figure that includes VIs of all the lags is moved to the supplements. We deeply understand the critics related to the soil moisture data. The results related to soil wetting are now discussed using the soil moisture, WTL and precipitation results together, instead of focusing mainly on the soil moisture. VI and ALE results of soil moisture are well in line with results related to WTL and precipitation. Therefore, instead of decreasing the amount of discussion related to soil wetting, we better supported the findings also with WTL and precipitation results. The problems and uncertainties related to soil moisture data are also brought up more openly in the material and methods section as well as in the discussion.

**About VIs and lag-times:**
The ability of VIs to show the importance of a certain lag is affected a) by the fact that lack-times can vary in time, b) lag-effects may not be always present (for example, only after soil wetting) and c) the effect depends on the values of other variables (interactions). In addition, the interpretation of VIs of correlated versions of the variable (unlagged and lagged) is difficult and sensitive to false interpretation (See response nr. 10). The fact that lags for soil moisture (7 cm and 20 cm) received increased VI scores tells about the overall importance of lags in responses of N2O flux to soil moisture.

The conclusions about the length of lag-times are made from ALE plots that enable comparing mean predicted fluxes between unlagged and different lagged variables for each environmental variable. The lag-times were interpreted from the ALE plots in the following way: Out of the unlagged and lagged versions of a variable, the one for which the model had predicted highest N2O flux compared to the mean prediction of the model (ALE value=0), was considered to best represent typical lag-time between environmental condition and peak flux. In the case of soil moisture, only soil moisture values higher than 0.35 $m^{-3}$ $m^{-3}$ were considered when interpreting lag-dynamics in response to precipitation events on thawed soil since very low soil moisture values on ALE plots tell about the response to freezing. We have included a short explanation about the identification of lag-times also in the methods section.

See lines:

258-268: *"VIs and accumulated local effects (ALE) were used to interpret the modeling results. For easier comparison of VIs across chambers, the VIs of each chamber were scaled from zero to one (0 = least important variable, 1 = most important variable) and the total VIs of each variable were calculated (total VI = VI of unlagged variable + VIs of lags). The ALE method by Apley and Zhu (2020) was used to visualize the response of N2O flux to environmental conditions and their lags in the models. In ALE figures, ALE value (y-axis) zero refers to the mean predicted N2O flux, with a positive ALE value meaning larger and a negative value lower predicted N2O flux in a specific environmental condition (x-axis). ALE values for lagged environmental variables indicate the response of predicted N2O flux to previous environmental conditions. From the unlagged and lagged versions of each environmental variable, the one that received the highest ALE value for a given environmental condition was considered to represent the typical response time of N2O flux to that condition. In this article, the response time, or lag length in the presence of at least a one-day lag, refers to the time it takes for N2O to reach peak flux after the onset of a given environmental condition."*

404-408: *"Figure 8: Total variable importance (VI) of different environmental variables in explaining the temporal variation of N2O flux in random forest with conditional inference trees. Total VI is the sum of VIs of unlagged and lagged (1–7 days) versions of the variable. Rows in the matrix plot show VIs for different chambers and the mean VIs across all the chambers (Mean). VI scores are scaled between 0 and 1 (0 = no importance, 1 = highest importance) per chamber. Lag-specific VIs are shown in Fig. S6."*

Reviewer (L409-410): I do not understand this sentence, 'and increased VI scores also for lagged variables with the mean across lags 0.25.' So far everything with lag decreased VI scores, so what does the 'also' refer to? And actually, in this case too, the VI score decreased from 0.27 to 0.25. Was that just a typo? Maybe write shorter sentences or use commas but I am not able to follow the information in this sentence.

**30.** Response: We reformatted the section to make it easier to read.

See lines 396-402: *"Soil moisture (both 7 and 20 cm), air temperature and WTL were considered to be the most important variables explaining the temporal variation of N2O flux (Fig. 8) with the mean total variable importance (VI, 0 = no importance, 1 = high importance) being 0.7 and 0.6 for soil moisture (7 and 20 cm respectively) and 0.5 for air temperature and WTL. The mean VI of lags (1–7 days) for each environmental variable was the highest for 7 and 20 cm soil moisture (mean VI 0.3 for both) with 5 cm soil temperature and air temperature also having importance on lags (mean VI 0.25 and 0.20, respectively, Fig. S6). Lags of other variables received mean VIs lower than 0.1, but precipitation had an increasing VI towards the longest lags (6–7 days)."*

Reviewer (L424): I am curious here, the ALE curves highlight that lagged moisture is more predictive when the moisture is already high. Why do you think that is? This is interesting and could be explored more. I haven't seen it in the discussion being mentioned again.

**31.** Response: ALE curves can be thought of as response curves with the predicted flux increasing towards the top of y-axis and values of environmental condition on x-axis. In our case, we have also added a curve for each lagged variable in the same plot. Interpretation goes like this: If, for example, 7-days lagged moisture variable gets the highest y-axis value on high moisture conditions, this tells that the model predicts high N2O flux when soil moisture was high 7 days

ago. If at the same time unlagged soil moisture gets the highest y-axis values on intermediate soil moisture conditions, we can interpret that the highest flux is predicted when soil moisture is on intermediate level but was high 7 days ago. High y-axis value for lagged variable on certain conditions does not tell about the predictive power of that variable on those conditions, although a variable that has a strong response to N2O is likely to have predictive power in the model. VI scores tell about the predictive power of each variable. The concepts and interpretation of VIs and ALEs are now explained better in the materials and methods section as well as in the figure captions to avoid confusion related to interpretation of VIs and ALEs.

See lines:
258-268: *"VIs and accumulated local effects (ALE) were used to interpret the modeling results. For easier comparison of VIs across chambers, the VIs of each chamber were scaled from zero to one (0 = least important variable, 1 = most important variable) and the total VIs of each variable were calculated (total VI = VI of unlagged variable + VIs of lags). The ALE method by Apley and Zhu (2020) was used to visualize the response of N2O flux to environmental conditions and their lags in the models. In ALE figures, ALE value (y-axis) zero refers to the mean predicted N2O flux, with a positive ALE value meaning larger and a negative value lower predicted N2O flux in a specific environmental condition (x-axis). ALE values for lagged environmental variables indicate the response of predicted N2O flux to previous environmental conditions. From the unlagged and lagged versions of each environmental variable, the one that received the highest ALE value for a given environmental condition was considered to represent the typical response time of N2O flux to that condition. In this article, the response time, or lag length in the presence of at least a one-day lag, refers to the time it takes for N2O to reach peak flux after the onset of a given environmental condition."*
404-408: *"Figure 8: Total variable importance (VI) of different environmental variables in explaining the temporal variation of N2O flux in random forest with conditional inference trees. Total VI is the sum of VIs of unlagged and lagged (1–7 days) versions of the variable. Rows in the matrix plot show VIs for different chambers and the mean VIs across all the chambers (Mean). VI scores are scaled between 0 and 1 (0 = no importance, 1 = highest importance) per chamber. Lag-specific VIs are shown in Fig. S6."*

Reviewer (L473-474): Maybe consider using other comparisons in the discussion chapter rather than numbers, e.g. the annual budget for three of the chambers was double in 2016 and 2017 compared to 2018. This makes it easier for the reader to follow your line of thought and to pick up your take-away message.

**32.** Response: We modified the discussion, and the original sentence was removed and replaced by different kind of sentence.

See lines 450-453: *"The N2O budgets of our six automatic chambers are unlikely able to represent the N2O budget of the whole site, but the mean annual N2O budget of the chamber area greater than 950 mg N2O m-2 y-1 in the two full study years out of three underlines the role of drained nutrient-rich peatland forest as hotspots for N2O emissions (Fig. 10)."*

Reviewer (L480): During harvest, were the roots left behind in the soil? The decomposition of finer and medium sized roots might fuel N2O emissions by substrate contribution. But of course, you are right, that doesn't explain the increase in the control site.

**33.** Response: Yes, the roots were left in the soil, although relatively little trees were cut close to the chambers. The effects of harvesting cannot be completely excluded but the fact that emissions increased similarly also in the control site after harvesting, imply that environmental conditions explain the increased fluxes in post-harvest years 2016-2017 (Korkiakoski et al., 2020).

See lines 456-460: *"Although the selection harvest done at the site in the spring 2016 did not increase the N2O budget of the harvested area compared to the control site according to Korkiakoski et al. (2020), the effect of the harvest on N2O fluxes of individual chambers cannot be completely excluded. Since the N2O budgets increased after harvesting in both the harvested and the control site (Korkiakoski et al., 2020), most of the increase in N2O budgets in the years 2016 and 2017 is likely explained by year-to-year variation in environmental conditions."*

Reviewer (L490): I agree, you should highlight more your findings of winter fluxes! More than a third of the annual budget was emitted during winter. That is worth stressing a bit more since many annual measurements do not cover this season.

**34.** Response: Thank you for the constructive feedback. Winter fluxes are an important part of the annual N2O budgets, and we made a decision to considerably expand the discussion related to winter fluxes.

See for example lines 463-474: *"Winters were characterized by N2O flux peaks occurring during both freezing and thawing (Fig. 6, 7 and S5), similar to those reported in earlier studies (Teepe et al., 2001; Maljanen et al., 2007; Maljanen et al., 2010). Freezing-related N2O emissions are likely explained by N2O production in the remaining unfrozen water films that have increased C and N content in the freezing soil (Maljanen et al., 2007; Congreves et al., 2018). Winter N2O flux peaks were measured when soil frost reached at least the 5 cm depth, whereas during winters with only shallow frost (< 5 cm, winters 2017–2018 and 2018–2019), high N2O fluxes were less common. This indicates the importance of frost depth for winter N2O emissions. The importance of ground frost severity and depth has also been suggested by others in several ecosystems (Nielsen et al., 2001; Koponen and Martikainen, 2004; Maljanen et al., 2007; Luo et al., 2012). The importance of deeper soil freezing may indicate that the freezing-related N2O fluxes mainly originate from the freezing peat rather than from the surface litter layer, unlike suggested by Pihlatie et al. (2007) in a nutrient-poor peatland forest. Low C:N ratio may have favored N2O production in the nutrient-rich peat (Klemedtsson et al., 2005). Site-specific differences in nutrient availability may influence the sensitivity of winter N2O fluxes to frost depth."*

Reviewer (L520-525): You could play around here way more with what you got from your results! These winters with discontinuous snow cover are increasing with processing climate change due to warmer temperatures. This trend will then also most likely increase N2O budgets. Highlight more and in the abstract too! On the other hand, summers are predicted to be drier too, for the same reason of rising temperatures. Would this balance the annual budgets? Have you thought of doing some calculations for this, as a potential outlook for your study site? Discuss more the bigger picture conclusion from your results.

**35.** Response: This is indeed an interesting point. We highlighted the conclusions related to climate change more in the discussion, conclusions and in the abstract. We however kept the main focus

on spatio-temporal variation of N2O and didn't do any further projections or calculations related to climate change.

See lines:
479-481: *"The results suggest the possibility for increasing winter N2O emissions from drained peat soils if winters continue to warm, the occurrence of extreme temperature fluctuations increases and snow cover in the southern boreal region becomes shallower."*
511-513: *"Our results on summer and winter N2O fluxes suggest that low N2O fluxes during dry summers might offset the effect of the increasing winter N2O fluxes on annual N2O budgets if dry summers become more frequent in the warming climate."*
599-603: *"Winter N2O emissions will likely increase in the future due to warming winters with shallow and discontinuous snow cover. Summer N2O emissions may decrease and possibly offset the effect of warming winters on annual N2O budgets in dry years. Year-to-year variation in N2O emissions will likely increase as extreme weather events are predicted to become more frequent."*

Reviewer (L528): No need to repeat yourself. Rather use an introductory sentence on the spatial focus now. The next sentence is sufficient.

**36.** Response: We removed the repetitive sentence.

Reviewer (L533): Could you remind me here of how big the difference was with a comparison of magnitude? E.g. on average double/triple the annual budget.

**37.** Response: This sentence was removed.

Reviewer (L591): Not even in the hourly resolution?

**38.** Response: Not at least in the scale that it could be spotted from the data and linked to thawing. When soil surface started to reach > 0°C temperatures during daytime in spring, the N2O flux started to increase slowly during the next weeks and months with some diurnal variation. Temporal variation in spring was strongly dominated by day-to-day or week-to-week variation rather than hourly variation. We added a sentence about the small importance of hourly variation in the materials and methods section.

158-159: *"Hourly N2O flux peaks were not typical in the flux data, and daily mean N2O fluxes thus well represent the main characteristics of the temporal variation."*

Reviewer (L615-632): Please change the focus. Your soil moisture is not fully representative of the chamber conditions and have no replicates! General conclusions here are inappropriate due to the limited amount of soil moisture measurements. However, precipitation shows a quite consistent gradual VI score increase towards the lag time of 7 days. Precipitation is less likely to vary on the micro-scale as much as soil moisture and has therefore, in my opinion, a stronger message here. Please consider revising throughout the manuscript.

**39.** Response: The lack of chamber-specific or close-to chambers soil moisture data is an unfortunate limitation in our study. The measured soil moisture is most likely able to describe the general temporal variation in soil moisture relatively well, but we cannot say anything about the absolute level of soil moisture in different chamber locations or inside chambers. We admit that varying peat properties, evapotranspiration patterns and tree cover can create variation also

in temporal dynamics of soil moisture, and these dynamics are not captured here. For example, part of the soil may have dried faster after rain due to variation in peat properties, which can affect the lag-dynamics seen in response to precipitation events and explain differences in lag-times between chambers. We have been careful with the conclusions made from soil moisture results alone and when conclusions are made, we have supported the findings with WTL and precipitation results that strongly support soil moisture results. We have also more openly brought up uncertainties related to soil moisture both in the materials and methods sections as well as in the discussion. We appreciate your critical comment.

See lines:
175-180: *"The soil moisture data were used to describe the temporal variation of soil moisture, assuming that the soil moisture had relatively similar temporal patterns across the study site. The absolute level of soil moisture in each chamber may have differed from the measured soil moisture, and the possibility of differences in the temporal variation of soil moisture between the logger and the chambers cannot be excluded. Soil moisture data were used together with water table level and precipitation data to strengthen the conclusions related to soil water conditions."*
561-562: *"Compared to the previous studies, the observed lag times are long, with indication for even longer lags than seven days in some chambers."*
576-577: *"To determine exact lag times in response to soil moisture peaks, chamber-specific soil moisture data would be required.."*

Reviewer (Appendix B L664): How was the RMSE used to analyze the prediction accuracy? This is all about the R2. A bit more context would be helpful to make the data processing as transparent as possible.

**40.** Response: RMSEs were used as an additional metric to compare model performance in different evaluation datasets of each chamber. Due to harder interpretation of RMSE compared to R2, R2 values are mainly discussed in the text. RMSE values depend on the values of the data, which makes comparing model performance based on RMSEs of different chambers difficult. Comparing RMSEs of within-training-period evaluation datasets (OOB and 30% of training period data) to RMSEs of outside-training-period evaluation data (fourth measurement year) is also made complex by the fact that the fluxes during the fourth year of measurements were lower than the fluxes during training period. RMSEs have not affected modelling process (for example, no variable selection), but have been used as an additional evaluation metric to provide information for interested readers. We added a sentence about this in the materials and methods section.

See lines 255-256: *"The performance of the models on different evaluation datasets was analyzed using R squared (R2) and root mean squared error (RMSE). R2 was used to compare model performance between chambers. Variable selection was not done."*

Technical comments:

Reviewer (L38): I would suggest using the GWP with feedback effect accounting for 298 CO2-eq.

**41.** Response: We updated the GWP according to the most recent IPCC report. The timespan of the GWP was also added.

See lines 38-40: *"Among the greenhouse gases whose emissions contribute to climate change, nitrous oxide (N2O) is one of the most potent, with a 100-year global warming potential 273 times greater than that of carbon dioxide (Forster et al., 2021)."*

Reviewer (L38-41): A major part of the N2O emissions of N2O originates from soils (Butterbach-Bahl et al., 2013; Davidson and Kanter, 2014) and human impact through altered nitrogen (N) cycle. Land use and climate change affect the soil N2O emissions both in natural and managed ecosystems (Tian et al., 2018, 2020).

**42.** Response: The sentence was split into two to make it easier to read.

See lines 40-42: *"A major part of the emissions of N2O originate from soils (Butterbach-Bahl et al., 2013; Davidson and Kanter, 2014). Human impact through altered nitrogen (N) cycle, land use and climate change affect the soil N2O emissions both in natural and managed ecosystems (Tian et al., 2018, 2020)."*

Reviewer (L424): Write ALE out, no abbreviations at the beginning of the sentence, and it is a good reminder here what it means for everyone not too familiar with it.

**43.** Response: Yes, we added the full word here with the abbreviation in parentheses.

See line 410: *"Accumulated local effects (ALE) for 7 cm soil moisture showed…"*

Reviewer (L591): Thawing. Ice melts, soil thaws.

**44.** Response: The correct verb is now used throughout the manuscript.

See lines 494-496: *"In the present study, spring soil thaw triggered N2O emissions, but emissions increased slowly with increasing soil temperature and peaked in late spring or summer, significantly later after soil thaw than reported in previous studies"*

References:

Apley, D. W., and Zhu, J.: Visualizing the effects of predictor variables in black box supervised learning models. Journal of the Royal Statistical Society Series B: Statistical Methodology, 82(4), 1059-1086, https://doi.org/10.1111/rssb.12377, 2020

Korkiakoski, M., Ojanen, P., Penttilä, T., Minkkinen, K., Sarkkola, S., Rainne, J., Laurila, T., and Lohila, A.: Impact of partial harvest on CH4 and N2O balances of a drained boreal peatland forest, Agric. For. Meteorol., 295, 108168, https://doi.org/10.1016/j.agrformet.2020.108168, 2020

Rautakoski, H., Korkiakoski, M., Aurela, M., Minkkinen, K., Ojanen, P., and Lohila, A.: 4.5 years of peatland forest N2O flux data data measured using automatic chambers, Zenodo, https://doi.org/10.5281/zenodo.8142188, 2023a

Rautakoski, H., Korkiakoski, M., Aurela, M., Minkkinen, K., Ojanen, P., and Lohila, A.: Supplementary material to the article "Exploring temporal and spatial variation of nitrous oxide flux

using several years of peatland forest automatic chamber data", Zenodo, https://doi.org/10.5281/zenodo.8141569, 2023b

Strobl, C., Boulesteix, A.-L., Zeileis, A., and Hothorn, T.: Bias in random forest variable importance measures: Illustrations, sources and a solution, BMC Bioinform., 8(1), 25, https://doi.org/10.1186/1471-2105-8-25, 2007

Strobl, C., Hothorn, T., & Zeileis, A. (2009). Party on!

**Referee 3:**

This paper explores the temporal variation in "elevated" N2O fluxes in a peatland forest using long-term automated flux data. Long-term flux data is needed to better understand the spatial and temporal dynamics of N2O fluxes, which are difficult to predict across space and time. Attempting to decipher such an extensive and variable dataset is no easy task, but I believe incorporating the comments from the other reviewers would help tease out the novelty of this dataset and how it improves our understanding. Given the extensive, constructive comments already provided by the other reviewers, I will try to keep my comments relatively succinct.

Other general comment (either to author or to editor): Not including all line numbers is particularly difficult to review. Having to count each line from five is much more tedious than dealing with the "extra" text from including all line numbers.

**45.** General response: We thank you for your comments. We have modified the manuscript to better highlight the important results and included also all the row numbers in the revised version.

Specific comments:

Reviewer (L 190-193): There certainly is a point to not only being interested in the "hottest" hot moments like those that you cite, but perhaps you can provide a sentence or too that succintly describes your method so that it's easy to cite/use in the future.

**46.** Response: We decided to leave the section as it was.

Reviewer (L199): However, it does seem that you are more-or-less simply just choosing "days above mean flux" as having high fluxes: "The mean N2O flux of the study period was close to the chosen 70 % percentile threshold in all chambers. Days with the mean flux above the 70 % percentile were classified as high-flux days." What inherent value is that besides being high?

**47.** Response: The values of chamber-specific means and 70 % percentiles are shown in Table 2. As the use of chamber-specific 70% percentile suggests, the definition of high-flux is here proportional to the range of the temporal variation measured in each chamber. In chambers with small fluxes and small range of flux variation, the days classified as high-flux periods can have fluxes that were considered low in other chambers.

Reviewer (L207,209): Not exhaustive, but N2O is not subscript on these lines. Please doublecheck throughout.

**48.** Response: We have now double-checked the text.

Reviewer (L610-626): This may likely be the case, but so far datasets collected can only infer this, could you potentially provide some advice on what measurements need to be taken in the future to further our understanding of this microsite variability. Can we predict this variability?

**49.** Response: Soil properties likely play a role here since they can determine the availability of substrate, soil oxygen conditions and drainage after rainfall. We added general advice to analyze the soil properties from sites where N2O fluxes are measured. To predict variability in lag-times, more information about the variability of lag-times in different ecosystems and soils is needed. With our data, we are only able to hypothesize possible linkages to environmental conditions and spatially varying soil properties.

See lines 543-546: *"The results emphasize the importance of comprehensive soil sampling (e.g. N forms, bulk density, pH, C:N, root density) and chamber-specific measurements of environmental variables (e.g. soil moisture, soil temperature, WTL), when studying spatio-temporal variation of N2O flux, especially in the forested study sites with variable microclimate."*

Reviewer (L643-645): While understanding the variability is important (and should be condensed into a less extensive part of the manuscript), I think this is one of the most important takeaways and should be expanded: how do these compare to other forest peatland measurements? Were less extensive measurements still producing similar mean flux values?

**50.** Response: We have now expanded the take-away message in the conclusions and abstract as the Reviewer 2 also suggested (See response nr. 35). Conclusion related to N2O budgets in the changing climate are also discussed more and discussion section includes concluding sentences.

See lines:
479-481: *"The results suggest the possibility for increasing winter N2O emissions from drained peat soils if winters continue to warm, the occurrence of extreme temperature fluctuations increases and snow cover in the southern boreal region becomes shallower."*
511-513: *"Our results on summer and winter N2O fluxes suggest that low N2O fluxes during dry summers might offset the effect of the increasing winter N2O fluxes on annual N2O budgets if dry summers become more frequent in the warming climate."*
597-603: *"The observed high N2O fluxes from the peatland forest highlight the role of nutrient-rich drained peat soils as hotspots for N2O emissions in the boreal region. The dependence of N2O budgets on seasonally varying weather conditions suggests high sensitivity of peatland forest N2O budgets to changing climate. Winter N2O emissions will likely increase in the future due to warming winters with shallow and discontinuous snow cover. Summer N2O emissions may decrease and possibly offset the effect of warming winters on annual N2O budgets in dry years. Year-to-year variation in N2O emissions will likely increase as extreme weather events are predicted to become more frequent."*